# Continued increase of CFC-113a (CCl₃CF₃) mixing ratios in the global atmosphere: emissions, occurrence and potential sources

Karina E. Adcock[1], Claire E. Reeves[1], Lauren J. Gooch[1], Emma C. Leedham Elvidge[1],

Matthew J. Ashfold[2], Carl A. M. Brenninkmeijer[3], Charles Chou[4], Paul J. Fraser[5],

Ray L. Langenfelds[5], Norfazrin Mohd Hanif[1], Simon O'Doherty[6], David E. Oram[1,7],

Chang-Feng Ou-Yang[8], Siew Moi Phang[9], Azizan Abu Samah[9], Thomas Röckmann[10],

William T. Sturges[1], and Johannes C. Laube[1]

[1]Centre for Ocean and Atmospheric Sciences, School of Environmental Sciences, University of East Anglia, Norwich, NR4 7TJ, UK
[2]School of Environmental and Geographical Sciences, University of Nottingham Malaysia Campus, 43500 Semenyih, Malaysia
[3]Air Chemistry Division, Max Planck Institute for Chemistry, Mainz, Germany
[4]Research Center for Environmental Changes, Academia Sinica, Taipei 11529, Taiwan
[5]Oceans and Atmosphere, Climate Science Centre, Commonwealth Scientific and Industrial Research Organisation, Aspendale, Australia
[6]Department of Chemistry, University of Bristol, Bristol, UK
[7]National Centre for Atmospheric Science, School of Environmental Sciences, University of East Anglia, Norwich, NR4 7TJ, UK
[8]Department of Atmospheric Sciences, National Central University, Taipei, Taiwan
[9]Institute of Ocean and Earth Sciences, University of Malaya, Kuala Lumpur, Malaysia
[10]Institute for Marine and Atmospheric Research Utrecht, Utrecht University, Utrecht, the Netherlands

*Correspondence to*: Karina Adcock (Karina.Adcock@uea.ac.uk)

## Abstract

Atmospheric measurements of the ozone depleting substance CFC-113a (CCl₃CF₃) are reported from ground-based stations in Australia, Taiwan, Malaysia and the United Kingdom, together with aircraft-based data for the upper troposphere and lower stratosphere. Building on previous work we find that, since the gas first appeared in the atmosphere in the 1960s, global CFC-113a mixing ratios have been increasing monotonically to the present day. Mixing ratios of CFC-113a have increased by 40 % (percent) from 0.50 to 0.70 ppt (parts per trillion) in the Southern Hemisphere between the end of the previously published record in December 2012 and February 2017. We derive updated global emissions of 1.7 Gg yr⁻¹ (gigagrams per year) on average between 2012 and 2016 using a two-dimensional model. We compare the long-term trends and emissions of CFC-113a to those of its structural isomer, CFC-113 (CClF₂CCl₂F), which still has much higher mixing ratios than CFC-113a, despite its mixing ratios and emissions decreasing since the 1990s. The continued presence of Northern Hemispheric emissions of CFC-113a is confirmed by our measurements of a persistent interhemispheric gradient in its mixing ratios, with higher mixing ratios in the Northern Hemisphere. The sources of CFC-113a are still unclear, but we present evidence that indicates large emissions in East Asia, most likely due to its use as a chemical involved in the production of hydrofluorocarbons. Our aircraft data confirm the interhemispheric gradient as well as showing mixing ratios consistent with ground-based observations and the relatively long atmospheric lifetime of CFC-113a. CFC-113a is the only known CFC for which abundances are still increasing substantially in the atmosphere.

## 1. Introduction

The ozone layer in the stratosphere blocks most of the harmful solar ultraviolet radiation from reaching the Earth's surface and therefore protects human health and the environment. Chlorofluorocarbons (CFCs) are industrially produced chemicals that were commonly used as refrigerants, aerosol propellants, solvents and foam blowing agents. CFCs have negligible loss mechanisms in the troposphere and only break down when they reach the stratosphere where they are exposed to strong ultraviolet light and decompose mostly through photolysis and reaction with $O^1D$ (Ko et al., 2013). These decomposition products act as catalysts in the destruction of ozone and they have, in combination with other chlorine and bromine containing gases, led to the formation of the ozone hole (Farman et al., 1985; Molina and Rowland, 1974). The discovery of this phenomenon motivated the `Montreal Protocol on Substances that Deplete the Ozone Layer', an international agreement to phase out the use of CFCs and other ozone depleting substances (ODSs) (UNEP, 2016a). It came into force in 1989 and, other than for a few critical use exceptions, there has been a global ban on CFC production since 2010 (UNEP, 2016a). Due to this, mixing ratios of most CFCs are now decreasing in the atmosphere and the ozone hole shows signs of recovery (Pawson et al., 2014; Solomon et al., 2016). Continued reductions in CFC mixing ratios are needed to allow the ozone layer to recover to pre-1970 levels.

Recently, mixing ratios of CFC-113a ($CCl_3CF_3$), the structural isomer of the well-known ozone-depleting substance CFC-113 ($CF_2ClCFCl_2$), were found to still be increasing in the atmosphere up until 2012 (Laube et al., 2014). The previously published evidence for increasing mixing ratios of CFC-113a comes from air samples collected at Cape Grim, Tasmania (41° S) and firn air data collected in Greenland (77° N) in 2008 (NEEM project) (Buizert et al., 2012; Laube et al., 2014). The firn air depth profile data, when combined with inverse modelling, provide smoothed time series of compound mixing ratios going back up to a century (Buizert et al., 2012; Laube et al., 2012). CFC-113a became detectable in the atmosphere in the 1960s (Laube et al., 2014). Cape Grim is a clean-air measurement site located in Tasmania, Australia, with air sampling/analysis activities since 1976 and the CFC-113a record derived from the Cape Grim Air Archive (1978 onwards) shows mixing ratios increasing over time with a sharp acceleration starting around 2010 (Laube et al., 2014). Global annual emissions of CFC-113a were estimated using a two-dimensional atmospheric chemistry-transport model, showing increases since the 1960s and more than doubling between 2010 and 2012, reaching 2.0 Gg yr$^{-1}$ in 2012 (Laube et al., 2014). In addition, measurements of aircraft samples from the CARIBIC-IAGOS observatory identified an interhemispheric gradient with mixing ratios increasing from the Southern Hemisphere to the Northern Hemisphere; and the atmospheric lifetime of CFC-113a was estimated at 51 years from stratospheric research aircraft flights in late 2009 and early 2010 (Laube et al., 2014).

The origin of the emissions that cause this increase in CFC-113a mixing ratios is as yet undetermined. Some evidence of a potential connection with hydrofluorocarbon (HFC) production has been found (Laube et al., 2014) and here we use additional data to investigate this possibility further. Laube et al. (2014) reported data until 2012. This study uses data that have become available since 2012 to provide an update on its global trend and emissions and to assess these in terms of our understanding of the sources of this gas and its potential impact on ozone.

## 2. Methods

### 2.1 Analytical technique

Air samples from all the campaigns discussed in this study were collected in electropolished and/or silco-treated stainless steel gas cylinders, except for the CARIBIC observatory, for which samples were collected using glass-bottle based samplers (Brenninkmeijer et al., 2007). Various pumps were used for the different sampling activities, all of which have been thoroughly tested for a large range of trace gases (Brenninkmeijer et al., 2007; Laube et al., 2010a; Allin et al., 2015 and Oram et al., 2017). After collection, the samples were transported to the University of East Anglia (UEA) to be analysed on a high-sensitivity gas chromatograph coupled to a Waters AutoSpec magnetic sector mass spectrometer (GC-MS). The trace gases were cryogenically extracted and pre-concentrated. A full description of this system can be found in Laube et al. (2010b). Analysis was partly carried out using a GS GasPro column (length ~50 m, ID 0.32 mm) and partly with a KCl-passivated $Al_2O_3$-PLOT column (length: 50 m, ID 0.32 mm), (Laube et al., 2016). The latter analysis has been slightly modified by the addition of

an Ascarite filter to remove carbon dioxide. Several tests and comparisons ensured that no significant differences in CFC-113 and CFC-113a mixing ratios were obtained regardless of the column or filter used. A possible interference could arise when measuring CFC-113a on the GS GasPro column using m/z 116.91 if concentrations of the nearby eluding HCFC-123 are high. This was the case for a small number of samples analysed for this work and those measurements were either a) repeated using the interference-free m/z 120.90, b) replaced with measurements on the KCl-passivated $Al_2O_3$-PLOT column, or c) excluded. The KCl-passivated $Al_2O_3$-PLOT column separated CFC-113 and CFC-113a well, no interferences were observed and m/z 116.91 was used for quantification. All the samples are compared to the same NOAA standard (AAL-071170) and there were routine measurements of multiple standards to exclude the possibility of mixing ratio changes in the standard over time. The samples collected in Taiwan in 2013 were also measured on the Entech-Agilent GC-MS system operating in electron ionisation (EI) mode. This consists of a preconcentration unit (Entech model 7100) connected to an Agilent 6890 GC and 5973 quadrupole MS (Leedham Elvidge et al., 2015). The calibration scale used for CFC-113a is the UEA calibration scale and for CFC-113 is the NOAA 2002 calibration scale. On a typical day, the working standard is measured five to eight times, between every two or three samples. The sample peak sizes are measured relative to the standards measured just before and after them. The working standard is used to correct for small changes in instrument response over the course of a day. The dry air mole fraction (mixing ratio) is measured and the units, parts per trillion (ppt) are used in this study as an equivalent to picomole per mole. The measurement uncertainties are calculated the same way for all measurements and represent one sigma standard deviation. They are based on the square root of the sum of the squared uncertainties from sample repeats and repeated measurements of the air standard on the same day.

**2.2 Sampling**

The following new data are presented in this study (see also Figure 1 and Table 1):

1. Laube et al. (2014) reported CFC-113a measurements from Cape Grim, Tasmania from 1978 to 2012. We now report four more years of CFC-113a measurements from Cape Grim, up to February 2017. From 2013 to 2017, 20 samples were collected at Cape Grim at irregular intervals of between one to five months apart. The CFC-113 mixing ratios (1978-2017) from analyses of archived air samples collected at Cape Grim, Tasmania and analysed at the UEA, together with NOAA flask data, and AGAGE *in situ* data are also included to compare the two isomers. CFC-113 stability in the Cape Grim Air Archive has been demonstrated in the AGAGE program for periods up to 15 years and longer (Fraser et al., 1996; CSIRO unpublished data). Most of the CFC-113 UEA Cape Grim data set was previously published in Laube et al. (2013. Some of the earlier samples from Laube et al. (2013) and Laube et al. (2014) were reanalysed on the KCl-passivated $Al_2O_3$-PLOT column (length: 50 m, ID 0.32 mm). They showed very good agreement with the previous GasPro column-based measurement with comparable precisions and no detectable offset. The Cape Grim air samples were collected under background conditions with winds from the south-west, marine sector, so that sampled air masses were not influenced by nearby terrestrial sources and are representative of the extra-tropical Southern Hemisphere. Details of the sampling procedure have been reported in previous publications (e.g. Fraser et al., 1999; Laube et al., 2013).

2. Tacolneston tower is a measurement site in Norfolk (Ganesan et al., 2015), and is part of the UK Tall Tower Network. Air samples were collected on a near-biweekly basis between July 2015 and March 2017 using an air inlet at 185 m.

3. Ground-based samples were collected from Bachok Marine Research Station on the northeast coast of Peninsular Malaysia in January and February 2014.

4. During the StratoClim campaign (http://www.stratoclim.org/), air samples were collected during two flights by the Geophysica high altitude research aircraft, as described in Kaiser et al. (2006), in the upper troposphere and lower stratosphere (10-20 km) over the Mediterranean on 01-Sep-2016 and 06-Sep-2016.

5. Air samples were collected at regular intervals at altitudes of 10-12 km during long distance flights on a commercial Lufthansa aircraft from 2009 to 2016 (Brenninkmeijer et al., 2007) on four flights between Frankfurt, Germany and Bangkok, Thailand; five flights between Frankfurt, Germany and Cape Town, South Africa; and one flight between Frankfurt, Germany and Johannesburg, South Africa; including the four flights referred to in Laube et al. (2014) (CARIBIC project, www.caribic-atmospheric.com).

6. Four ground-based air sampling campaigns took place in Taiwan from 2013 to 2016. Between 19 and 33 air samples were collected in March and April each year. In 2013 and 2015 samples were collected from a site on the southern coast of Taiwan (Hengchun) and in 2014 and 2016 samples were collected from a site on the northern coast of Taiwan (Cape Fuguei). See also Vollmer et al. (2015), Laube et al. (2016) and Oram et al.
145    (2017).

## 2.3 Emission modelling

A two-dimensional atmospheric chemistry-transport model was used to estimate, top-down, global annual emissions of CFC-113a and CFC-113 for the purpose of comparing the emissions of the two isomers. The model contains 12 horizontal layers each representing 2 km of the atmosphere and 24 equal-area zonally averaged
latitudinal bands. The modelled mixing ratios for the latitude band that Cape Grim is located within (35.7° S–41.8° S) were matched as closely as possible to the observations at Cape Grim (40.7° S) by iteratively adjusting the global emissions rate until the differences between the modelled mixing ratios and the observations were minimised. For more details about the model see Newland et al. (2013); and Laube et al. (2016).

This model was previously used to estimate the global annual emissions of CFC-113a (Laube et al., 2014). We
now update the CFC-113a emission estimates using an additional four years of Cape Grim measurements. The CFC-113 emissions are estimated using CFC-113 mixing ratios at Cape Grim for 1978-2017 from the UEA Cape Grim dataset and compared with bottom-up emissions estimates from the Alternative Fluorocarbons Environmental Acceptability Study (AFEAS, https://agage.mit.edu/data/afeas-data). The upper and lower emission uncertainties for CFC-113a and CFC-113 were determined by first calculating the uncertainty in
matching the modelled mixing ratios with the observed mixing ratios using their recommended atmospheric lifetimes and secondly considering the uncertainty range in the lifetimes. The 'best fit' (minimum-maximum) steady-state lifetimes used in this study are 51 years (30 years-148 years) for CFC-113a and 93 years (82 years-109 years) for CFC-113 (Ko et al., 2013; Leedham Elvidge et al., accepted ACP). Further details are provided in the supplementary material.

A latitudinal distribution of emissions, with 95 % of emissions originating in the Northern Hemisphere, was assumed for both compounds. As Cape Grim is a remote Southern Hemispheric site the emission distribution within the Northern Hemisphere has almost no effect on the modelled mixing ratios in the latitudinal band of Cape Grim. The emission distribution used for CFC-113 was assumed to be constant for the whole of the model run and has been used in previous studies for similar compounds (McCulloch et al., 1994; Reeves et al., 2005; Laube
et al., 2014, 2016). For CFC-113a we decided to select an emission distribution based on how well the modelled mixing ratios in the latitude band 48.6-56.4° N agreed with the observations at Tacolneston for the later part of the trend. Tacolneston can be considered to be representative of Northern Hemisphere background mixing ratios of CFC-113a for that latitude as there are no significant enhancements in mixing ratios (Figure 2). The emission distribution used in the CFC-113a model is the same as CFC-113 for the first 60 years (1934-1993) and then
gradually shifts over the next 10 years from more northerly latitudes (36-57° N) to more southerly latitudes (19-36° N). It then remains at more southerly latitudes until the end of the run in 2017. This distribution shift is based on the assumption that CFC-113a emissions are predominantly from Europe and North America at the beginning of the model run and then shift to be coming predominantly from East Asia towards the end of the model run. There are significant enhancements in CFC-113a mixing ratios in our measurements from Taiwan indicating
continued emissions in this region (Section 3.2.1) which is consistent with emissions in this latitude band in the model. The latter is also consistent with previous work that has found emissions of ozone depleting substances shifted from more northerly Northern Hemisphere latitudes to more southerly Northern Hemisphere latitudes (Reeves et al., 2005; Montzka et al., 2009). This is likely due to developing countries, which are mostly located further south, having more time to phase out the use of many ODSs than developed countries (Newland et al.,
2013; CTOC, 2014; Fang et al., 2016). With this emissions distribution, the modelled CFC-113a mixing ratios at Tacolneston matched closely to the observations (Figure 2). It should be noted that while there is evidence that supports the emission distribution used here, there might be alternative distributions that result in equally good fits to the trends, particularly in the earlier part of the record.

## 2.4 Dispersion modelling

The UK Met Office's Numerical Atmospheric Modelling Environment (NAME, Jones et al. 2007), a Lagrangian particle dispersion model, was used to produce footprints of where the air sampled during the Taiwan and Malaysia campaigns (Table 1) had previously been close to the Earth's surface. The model setup related to samples collected in Taiwan in 2016 was slightly different to the setup for simulations in 2013-2015; hereafter those differences are noted in parentheses, though they have no practical implications for our findings. The footprints were calculated over 12 days by releasing batches of 60,000 (30,000 in 2016) inert backward trajectories over a 3 hour period encompassing each sample. Over the course of the 12 day travel time the location of all trajectories within the lowest 100m of the model atmosphere was recorded every 15 minutes on a grid with a resolution of 0.5625° longitude and 0.375° latitude (0.25° by 0.25° in 2016). The trajectories were calculated using three-dimensional meteorological fields produced by the UK Met Office's Numerical Weather Prediction tool, the Unified Model (UM). These fields have a horizontal grid resolution of 0.35° longitude by 0.23° latitude for the 2013 and 2014 simulations, and 0.23° longitude by 0.16° latitude for the 2015 and 2016 simulations. In all cases the meteorological fields have 59 vertical levels below ~30km. Dates in the NAME footprint maps are presented in the format yyyy-mm-dd and use UTC time.

## 3. Results

### 3.1 Long-term atmospheric trends and estimated global annual emissions of CFC-113a and CFC-113

CFC-113a mixing ratios at Cape Grim were previously found to have been increasing from 1978-2012 (Laube et al., 2014, Figure 3). Since 2012, they have continued to increase from 0.50 ppt in December 2012 to 0.70 ppt in February 2017 (Figure 3). Between 1978 and 2009 the average rate of increase was 0.012 ppt yr$^{-1}$; between 2010 and 2017 the rate has risen threefold to about 0.037 ppt yr$^{-1}$ .

Although measurements at Tacolneston were made for a shorter time period (20 months), it also experienced an increase in CFC-113a mixing ratios of 0.03 ppt yr$^{-1}$ over the period July 2015 to March 2017, based on start and end points (Figure 2). Furthermore, for the CARIBIC flights the mean mixing ratios of CFC-113a increased on average, by 0.04 ppt yr$^{-1}$ between 2009 and 2016. Overall, there is a consistent picture of a continued global increase in background mixing ratios of CFC-113a. Its atmospheric burden has been increasing since the 1960s (Laube et al., 2014) and this has continued until early 2017, implying that ongoing emissions of CFC-113a exceed its rate of removal. The modelled global annual CFC-113a emissions began in the 1960s and increased steadily at an average rate of 0.02 Gg yr$^{-1}$ yr$^{-1}$ until they reached 0.9 Gg yr$^{-1}$ (0.6-1.2 Gg yr$^{-1}$) in 2010 followed by a sharp increase to 0.52 Gg yr$^{-1}$ yr$^{-1}$ from 2010 to 2012 when emissions were 1.9 Gg yr$^{-1}$ (1.5-2.4 Gg yr$^{-1}$) (Figure 3). We find that between 2012 and 2016, modelled emissions were on average 1.7 Gg yr$^{-1}$. The best model fit (minimum-maximum) suggests some minor and statistically non-significant variability between 1.6 Gg yr$^{-1}$ (1.3-2.0 Gg yr$^{-1}$) in 2015 and 1.9 Gg yr$^{-1}$ (1.5-2.4 Gg yr$^{-1}$) in 2012. See the supplementary material for more details.

It is instructive to look at CFC-113 to learn more about CFC-113a. The atmospheric trends of CFC-113 at Cape Grim (Figure 4) and estimated emissions are very different from those of CFC-113a. Mixing ratios of both compounds increased at the beginning of the record, but then the CFC-113 mixing ratios stabilised in the early 1990s and started to decrease (Figure 4), consistent with previous observations (Fraser et al., 1996; Montzka et al., 1999; Rigby et al., 2013; Carpenter and Reimann, 2014). This trend is similar to those of many other CFCs in the atmosphere (for example CFC-11 and CFC-12, Rigby et al., 2013), but in contrast to the increasing mixing ratios of CFC-113a. Note that CFC-113a mixing ratios are still much lower than those of CFC-113 even at the end of our current record in early 2017.  CFC-113 is the third most abundant CFC in the atmosphere (Carpenter and Reimann, 2014) and mixing ratios of CFC-113a are only about 1 % of CFC-113 mixing ratios in 2017. CFC-113 mixing ratios at Cape Grim measured by NOAA (https://www.esrl.noaa.gov/gmd/dv/ftpdata.html) and AGAGE (http://agage.eas.gatech.edu/data_archive/agage/) are also included in Fig. 4. There is a small offset of 2 % between the NOAA data and the current UEA Cape Grim dataset, with the UEA Cape Grim dataset being slightly higher, similar to the offset reported previously (Laube et al., 2013).

The CFC-113 model derived emissions begin in the 1940s and rapidly increase until they peak in 1989 at 252 Gg yr$^{-1}$, after which they decrease to 2.4 Gg yr$^{-1}$ in 2016 (Figure 4). This sharp decline witnesses the success of the Montreal Protocol, which came into force in 1989 and phased out the production of CFCs by 1996 in developed countries and 2010 in developing countries (UNEP, 2016a). The total cumulative emissions of CFC-113, up to the end of 2016, are 3164 Gg while the cumulative emissions of CFC-113a are 29 Gg, making the total cumulative emissions of CFC-113a less than 1 % of its isomer CFC-113. Alternatively, in the last decade, 2007-2016, cumulative emissions of CFC-113 are 38 Gg, while for CFC-113a they are 13 Gg, or a third of the CFC-113 cumulative emissions. Current CFC-113a emissions are similar to those of CFC-113 and could even surpass them if the trends continue (Figure 5).

Up until 1992, the CFC-113 emissions used in the model are the bottom-up emissions estimates from AFEAS. In the model, these emissions lead to a best-fit match to the CFC-113 observations. This shows that in the first part of the record, AFEAS report data accurately reflecting global CFC-113 emissions. However, after 1992 the AFEAS emissions lead to lower modelled mixing ratios than the observations indicating that AFEAS was missing some emissions after 1992. Therefore, the emissions used in our study here are the AFEAS emissions up until 1992. From 1992 onwards they are based on the best model fit to the UEA Cape Grim observations. CFC-113 emissions were also derived in another study using a range of emission inventories and estimates (Rigby et al., 2013). Those emissions mostly agree with ours within the uncertainties. Differences are likely due to this study using different lifetimes than Rigby et al. (2013).

The upper and lower bounds of the CFC-113 emissions in this study are derived using the 'likely' range in the CFC-113 lifetime given by SPARC of 82-109 years (Ko et al., 2013). The 'possible' range of 69-138 years was also estimated by Ko et al., (2013), however using a lifetime of 138 years the modelled mixing ratios did not decrease sufficiently rapidly after 1990 to match the observed downwards trend in CFC-113 even in the absence of emissions. We can use the observed decrease in CFC-113 mixing ratios from 2003 onwards to calculate a decay time (lifetime at zero emissions). For long lived gases with stratospheric sinks, such as CFC-113, the decay time and steady state lifetime are very similar differing by no more than 2 % (Ko et al., 2013). Setting the emissions to zero from 2003 onwards and adjusting the lifetime so that the model reproduces the CFC-113 mixing ratios at Cape Grim, suggests the lifetime for CFC-113 is 110 years. By assuming zero emissions, this lifetime is a maximum value, since any source of CFC-113 would have to be balanced by a shorter lifetime. Combining the measurement and model errors as described in the supplementary material gives an error of 5.7 %. Accounting for the 2 % error introduced by assuming the decay time is the same as the steady state lifetime gives are overall error of 6 %. Applying this to the lifetime gives a maximum lifetime of $110 \pm 7$ years. For comparison, we also calculated the maximum lifetime from the observed rate of decrease in CFC-113 mixing ratios at Cape Grim between 2003 and 2017 using the continuity equation and assuming no sources of CFC-113 (Supplementary material, Section 2). The agreement was good giving a maximum lifetime of $113 \pm 5$ years. It should be noted that CFC-113 is not the focus of this study, but we do find that emissions of it persist until 2017, which leaves room for the possibility that some of the recent emissions of CFC-113a are related to CFC-113 emissions, possibly through HFC production or agrochemical production (see Section 4) similar to findings for other isomeric CFCs (Laube et al., 2016).

### 3.2 Global distributions of CFC-113a

### 3.2.1 Enhancement above background mixing ratios

Many of the CFC-113a mixing ratios observed in Taiwan (light blue stars, Figure 6) are significantly higher than at the other locations considered in this study. The background mixing ratios consistently increase through this period from about 0.4 to about 0.7 ppt whereas the highest Taiwan samples have mixing ratios of up to 3 ppt. These enhancements in mixing ratios in all four years of the Taiwan campaigns indicate continued emissions in this region, most likely continental East Asia.

To determine the region(s) of emissions more accurately NAME footprints were used (Figure 7a-g). In general, when there are enhancements in CFC-113a mixing ratios then the NAME footprints usually show that the air most

likely came from the boundary layer over eastern China or the Korean Peninsula as shown in (a), (c), (d), and (g) for example. In contrast, the footprints in (b), (e) and (f) are examples of samples with lower CFC-113a mixing ratios and correspondingly there is very little influence from eastern China or the Korean Peninsula. However, we recognise the limitations of our relatively sparse dataset which prevents us from pinpointing the source region(s) further.

The mixing ratios in Taiwan are very variable indicating nearby source region(s) whereas Cape Grim and Tacolneston mixing ratios are much less variable. Therefore, the Taiwan measurements are better suited to investigate correlations that might shed further light on potential sources. After investigating correlations of CFC-113a with over 50 other halocarbons in samples from Taiwan we found CFC-113a mixing ratios correlate well ($R^2$>0.750) in multiple years with those of CFC-113 and HCFC-133a ($CH_2ClCF_3$) indicating a possible link between the sources of these compounds (Table 2). There is a great deal of variability in mixing ratios in the Taiwan samples. CFC-113a correlates well with CFC-113 in 2013 and 2014, but shows almost no correlation in 2015 and a slightly decreased correlation coefficient in 2016 (Table 2, Figure 8). In contrast, HCFC-133a strongly correlates with CFC-113a in the first three years (Table 2). The tropospheric lifetime of HCFC-133a is 4-5 years (McGillen et al., 2015) and its mixing ratios have varied in recent years. They increased in 2012/2013 and decreased in 2014/2015 (Vollmer et al., 2015). According to our latest data from Cape Grim, in 2016 they began increasing again. Large changes in emissions are needed to produce such a variable trend but the causes of these changes are still unclear (Vollmer et al., 2015).

CFC-113a mixing ratios in many of the samples collected at Bachok, Malaysia (grey crosses, Figure 6) are also enhanced above background levels although not to the same degree as the Taiwan samples, they range from 0.68 ppt to 1.00 ppt. The higher mixing ratios also have their origin in East Asian air masses being transported rapidly to the tropics by the East Asian winter monsoon circulation (Ashfold et al., 2015; Oram et al., 2017). Figure 9 shows an example NAME footprint from a sample collected in January 2014 that is representative for many other events.

The Tacolneston samples (yellow diamonds, Figure 6) show no significant enhancements in CFC-113a mixing ratios. This indicates the absence of regional sources in this part of the UK. Due to this and the relatively long lifetime of CFC-113a Tacolneston can be considered to be representative of Northern Hemisphere background mixing ratios of CFC-113a for that latitude. Both sites in Taiwan and also Tacolneston are Northern Hemisphere sites and although the Taiwan sites have many enhancements in CFC-113a mixing ratios there are some samples with background mixing ratios. For example, in spring 2016, the only period for which these datasets overlap, the lowest CFC-113a mixing ratio in Taiwan is 0.70 ppt on 24-Mar-2016 (Figure 7e). The closest Tacolneston sample to this is on 04-Apr-2016 with a CFC-113a mixing ratio of 0.71 ppt. This shows that Taiwan can encounter mixing ratios at background levels of CFC-113a. However, many of the air samples collected in Taiwan show mixing ratios of CFC-113a above background levels, indicating that enhanced levels of CFC-113a are generally widespread across this region.

### 3.2.2 Interhemispheric gradient of CFC-113a

For the period when measurements were made at both Cape Grim and Tacolneston (from July 2015 to February 2017), the Tacolneston mixing ratios were almost exclusively higher (though often indistinguishable within uncertainties) than the Cape Grim mixing ratios (Figure 6-inset). On average Cape Grim mixing ratios are 0.055 ± 0.024 ppt lower than Tacolneston mixing ratios. This shows that there is an interhemispheric gradient with higher CFC-113a mixing ratios in the Northern Hemisphere as would be expected for a compound emitted primarily in the Northern Hemisphere. This gradient is further supported by data from the six CARIBIC flights between Germany and South Africa for 2009-2016. The CARIBIC samples (purple circles, Figure 6) from the 2016 flight coincide temporally with the Tacolneston and the Cape Grim samples in Fig. 6 and confirm the observation of higher mixing ratios in the Northern Hemisphere (filled purple circles) and lower mixing ratios in the Southern Hemisphere (unfilled purple circles). Also see Fig. S1a in the supplementary material.

Laube et al. (2014) already found an interhemispheric gradient in CFC-113a using four of these CARIBIC flights 2009-2011 and furthermore discovered that the increasing trend of CFC-113a at Cape Grim, lagged behind the

increasing trend inferred from the firn air samples, collected to a depth of 76 metres, from Greenland, in the Northern Hemisphere. As the firn air measurements in the Laube et al. (2014) study were collected in Greenland between 14-30 July 2008, the surface measurements will be representative of atmospheric mixing ratios at that time. They will also be representative of background Northern Hemispheric CFC-113a mixing ratios for that latitude as the Greenland firn air location was isolated from any large industrial areas with potential sources of CFC-113a. Figure 6 includes the three measurements closest to the surface (brown crosses) although they are so close together that they appear to be one cross in the Figure and the average mixing ratio of the three samples is $0.44 \pm 0.01$ ppt.

Overall, these measurements demonstrate that there is an interhemispheric gradient in CFC-113a with higher mixing ratios in the Northern Hemisphere. This persistent interhemispheric difference indicates ongoing emissions of CFC-113a in the Northern Hemisphere with higher emissions in the Northern Hemisphere compared to the Southern Hemisphere. Similar interhemispheric gradients have been found for other CFCs (Liang et al., 2008), as CFCs are almost exclusively produced by industrial processes and most industrial production (and consumption) takes place in the Northern Hemisphere.

### 3.2.3 Measurements of CFC-113a in the stratosphere

Nearly all air samples collected during CARIBIC flights represent cruising altitudes of 10-12 km, which for samples over northern India, during four flights going from Germany to Thailand (green diamonds, Figure 6) would be near the tropopause. Their mixing ratios should be representative for air masses prior to entering the tropical tropopause region which is the main entrance region to the stratosphere (Fueglistaler et al., 2009). For the flight on 9-Nov-2013, there is some enhancement above background mixing ratios over South-East Asia (Figures 6, S1b). We speculate that this is likely due to air being transported from East Asia into the tropics via cold surges and then being transported up into the upper troposphere via convection (Oram et al., 2017). This means that the uplift mechanism in this region could potentially enhance concentrations of long-lived ODSs entering the stratosphere as compared to the 'background' clean air ground-based abundances that are normally used to derive such inputs (Carpenter and Reimann, 2014). The mechanism has already been proven to exist for shorter-lived gases (Oram et al., 2017) and we see very similar patterns transporting elevated mixing ratios of CFC-113a to the tropics very rapidly (within days) during a time of increased convective uplift.

The Geophysica flights reach altitudes of 20 km and so sample lower stratospheric air. The Geophysica 2009-2010 flights (pink squares) and the Geophysica 2016 flights (orange squares) begin at background mixing ratios and then decrease (Figure 6). During the 2016 flights, for example, measurements start at 10 km altitude where mixing ratios are 0.71 ppt and go up to 20 km where the mixing ratios are 0.36 ppt. In comparison to this, ground level measurements made at the Northern Hemisphere site, Tacolneston, had an average CFC-113a mixing ratio in 2016 of 0.72 ppt. In general, mixing ratios decrease as the aircraft ascends, mainly because air at higher altitudes will have taken longer to travel there and therefore is older and CFC-113a at higher altitudes has experienced photolytic decomposition. For more information about the Geophysica flights see the supplementary material.

### 4. Possible sources of CFC-113a

CFCs are entirely anthropogenic in origin. This means that there are processes either producing or involving CFC-113a that lead to continuing emissions of substantial amounts of this compound, especially in East Asia. Whilst the Montreal Protocol has banned the production and consumption of CFCs, there are exemptions including the use of ODSs as chemical feedstocks, chemical intermediates and fugitive emissions (UNEP, 2016a). As the Montreal Protocol does not require isomers to be reported separately, CFC-113 and CFC-113a may be reported together.

The strong correlations of CFC-113a with CFC-113 and HCFC-133a in Taiwan (Section 3.2.1) suggest that they are involved in the same production pathways or that their production facilities are co-located. There is an absence of a correlation between CFC-113a and CFC-113 in 2015 in Taiwan and in addition, the overall mixing ratios in 2015 appear to be lower than in the other years and have fewer large enhancements (Figure 8). This could be because in general less air was arriving from China/Korea in 2015, which is indicated by the NAME footprints

(Supplementary material, Section 5). Regions in China and Korea we found to be the most likely locations of CFC-113a emissions. Alternatively, the varying correlations in different years between CFC-113a and CFC-113 could be an indication of two or more independent sources of CFC-113a. CFC-113 feedstock use decreased by over 50 % in 2015 due to one producer, which is also a user choosing not to produce CFC-113 in 2015 and reducing in-house inventories instead (Maranion et al., 2017). If this were the process leading to correlated emissions of CFC-113a and CFC-113 it may explain their lack of correlation in 2015.

One possible source of CFC-113a is from HFC production, specifically, of HFC-134a ($CH_2FCF_3$) and HFC-125 ($CF_3CHF_2$), as both may involve CFC-113a in their production process. One of the pathways for production of HFC-134a begins with CFC-113 being isomerised to form CFC-113a, which is then fluorinated to produce CFC-114a ($CF_3CCl_2F$), the latter is then hydrogenated to produce HFC-134a (Manzer, 1990; Rao et al., 1992; Bozorgzadeh et al., 2001; Maranion et al., 2017). Another production method involves the reaction of hydrogen fluoride with trichloroethylene to form HCFC-133a and HFC-134a (Manzer, 1990; McCulloch and Lindley, 2003; Shanthan Rao et al., 2015). The process for manufacturing HFC-125 involves the starting materials of either HCFC-123 or HCFC-124. CFC-113a, CFC-113 and HCFC-133a can be formed as by-products when HCFC-123 and HCFC-124 are fluorinated and recycled during the process that forms HFC-125 (Kono et al., 2002; Takahashi et al., 2002).

If there were leaks in the system or venting of gases was practiced during these processes, this could lead to enhanced mixing ratios of CFC-113a and strong correlations with its isomer CFC-113 and HCFC-133a. HFC production should be contained and not involve fugitive emissions to the atmosphere. However, the Chemicals Technical Options Committee (CTOC) 2014 report suggests there may be small leaks, depending on the quality of the system, ranging between 0.1 % and 5 % of the feedstock used. The CTOC reported that a leak rate of about 1.6 % would be needed if all CFC-113a and HCFC-133a in the atmosphere had come from their use as feedstock in the production of HFC-134a, HFC-125 and HFC-143a, which is within the previous range (CTOC, 2014). HFC-143a is produced using HCFC-133a so it was included in the CTOC estimate but CFC-113a is not involved in its production so it is not included in this study (CTOC, 2014).

HFC-134a and HFC-125 mixing ratios are not well correlated with those of CFC-113a, CFC-113 or HCFC-133a, except for HFC-125 in 2016 that has a good correlation with CFC-113a (Table 2). We would not necessarily expect them to be well correlated as most of the emissions of the HFCs are usually related to their uses rather than their production. CFC-114a is also part of the production process of HFC-134a (Manzer, 1990), and can be another by-product during HFC-125 production (Kono et al., 2002; Takahashi et al., 2002). CFC-114a was only measured in 2015 and 2016 in Taiwan and was strongly correlated with CFC-113a in 2015 but not in 2016. This inconsistent correlation does not help to define further the source of CFC-113a. Furthermore HCFC-123 mixing ratios are not well correlated with CFC-113a, CFC-113 or HCFC-133a in any year in Taiwan but HCFC-124 mixing ratios are well correlated in 2015 with CFC-113a (Table 2) and with HCFC-133a ($R^2$=0.791). This strong correlation with HCFC-124 points to HFC-125 production being the dominant source in 2015.

As discussed above, eastern China and/or the Korean Peninsula are the most likely source regions for the elevated mixing ratios of CFC-113a observed in Taiwan, and the HFC industry in China has been growing rapidly in recent years (Fang et al., 2016). In China in 2013, productions of 118 Gg $yr^{-1}$ of HFC-134a and 78 Gg $yr^{-1}$ of HFC-125 were reported (Fang et al., 2016). Most industry in China is located on the eastern coast and the majority of HFC manufacturers are in the three eastern provinces of Shanghai, Zhejiang and Jiangsu. There are also HFC-134a and HFC-125 production plants in Japan, South Korea and Taiwan but the majority are located in China. The HFC production plants located in Taiwan could influence the mixing ratios at both the sites in Taiwan which introduces an additional uncertainty.

Alternatively, there is an official exemption in the Montreal Protocol for the use of CFC-113a as an "agrochemical intermediate for the manufacture of synthetic pyrethroids", (UNEP, 2003) probably because it is used to make the insecticides cyhalothrin and tefluthrin ( Brown et al., 1994; Jackson et al., 2001; Cuzzato and Bragante, 2002). In addition CFC-113 is a feedstock used to make trifluoroacetic acid (TFA) and pesticides (Maranion et al., 2017). CFC-113a is an intermediate in this process and these production processes are used in India and China and so this could also be a source in this region (Maranion et al., 2017). Furthermore HCFC-133a is also used to

manufacture TFA and agrochemicals although the process involving HCFC-133a is not related to the process involving CFC-113a (Rüdiger et al., 2002; Maranion et al., 2017).

Furthermore, CFC-113a is potentially present as an impurity in CFC-113 and the emissions of CFC-113a could be from CFC-113 banks. We saw in Sect. 3.2 that estimated emissions of CFC-113a began in the 1960s and HFC production did not become a large-scale industry until much later, so there must have been another source of CFC-
113a during that earlier part of the record. In Sect. 3.1 we concluded that there was possibly a small amount of continued emissions of CFC-113 to maintain the observed atmospheric mixing ratios. This would be consistent with either a source from banks and/or release in conjunction with CFC-113a.

To summarise we have identified four possible sources of CFC-113a: agrochemical production, HFC-134a production, HFC-125 production and an impurity in CFC-113. The correlations indicate that HFC production is
the dominant source in the East Asian region however; there is currently insufficient data available to conclude this with high confidence. Overall, the sources of CFC-113a emissions are still uncertain and further evidence is needed to quantify and pinpoint them. However, the likely sources we have found do not necessarily indicate a breach of the treaty as the use of CFCs as intermediates in the production of other compounds are permitted under the Montreal Protocol.

**5.  Conclusions**

There is a continued global increasing trend in CFC-113a mixing ratios based on a number of globally distributed sampling activities giving a consistent picture. CFC-113a mixing ratios at Cape Grim, Australia increased since the previous study from 0.50 ppt in December 2012 to 0.70 ppt in February 2017. The derived emissions were still significantly above 2010 levels and were on average 1.7 Gg yr$^{-1}$ (1.3-2.4 Gg yr$^{-1}$) between 2012 and 2016.
Additionally, CFC-113a mixing ratios vary globally and our findings confirm an interhemispheric gradient with mixing ratios decreasing from the Northern Hemisphere to the Southern Hemisphere. No significant emissions of CFC-113a occur in the UK but strong sources exist in East Asia. There are multiple possible sources of CFC-113a emissions and correlation analysis suggests the emissions might be associated with the production of HFC-134a and HFC-125.

The background abundances of CFC-113a reported here are currently small (<1.0 ppt) in comparison to the most common CFC, CFC-12 which has declining atmospheric mixing ratios of ~510 ppt in 2017 (NOAA, 2017). Therefore, the contribution of CFC-113a to stratospheric ozone depletion is comparably small and is not a cause for concern. While its increase in recent years has been considerable in percentage terms, it would have to continue increasing at this rate for several centuries before it reaches the atmospheric mixing ratios of the major CFCs in
the 1990s. For example, a constant emission of 2 Gg yr$^{-1}$ for CFC-113a yields a steady-state global mixing ratio of about 3.2 ppt. In 2016, HFCs were added to the Montreal Protocol and under the new amendment HFC consumption will be phased down in the coming decades (UNEP, 2016b). Therefore, if this phase down schedule is successful and the main source of CFC-113a is indeed from HFC production, then CFC-113a atmospheric mixing ratios should stop increasing in the future. However, whilst it seems likely, it is still not clear whether HFC
production is actually the main source of global CFC-113a emissions and whilst CFC-113a emissions have appeared to be stable in recent years this does not mean that they will not increase in the future. Further investigation and continued monitoring is needed to assess future changes and ensure the continued effectiveness of the Montreal Protocol. When continuous measurements of CFC-113a in the East Asia region become available the magnitude and origins of East Asian CFC-113a emissions can be quantified.

In the past, it was assumed that isomers of CFCs had similar uses, sources and trends and therefore it was not necessary to report them separately. However, in this study, we have found that the isomers CFC-113a and CFC-113 continue to have different trends in the atmosphere and in their emissions. Recently CFC-114a ($CF_3CCl_2F$) and CFC-114 ($CClF_2CClF_2$) were also found to have different trends and sources (Laube et al., 2016). If policy-makers wish to limit the impacts of individual isomers, then  atmospheric observational data on individual CFC
isomers should be reported to UNEP wherever possible. In addition, the increase in CFC-113a demonstrates that the use of ODSs as chemical feedstock or intermediates is becoming relatively more important as the use of ODSs

for direct applications decreases. If policy-makers target zero emissions of CFCs, then they might consider regulating these uses of ODSs.

## 6. Data availability

All data have been made publicly available in the supplement.

*Competing interests.* The authors declare that they have no conflict of interest.

## Acknowledgements

We are grateful for the work of the Geophysica team, the CARIBIC team (CARIBIC-IAGOS), the staff at the Cape Grim station, the NOAA Global Monitoring Division, and the AGAGE network. The StratoClim flights
were funded by the European Commission (FP7 project Stratoclim-603557, www.stratoclim.org). The collection and curation of the Cape Grim Air Archive is jointly funded by CSIRO, the Bureau of Meteorology (BoM) and Refrigerant Reclaim Australia.; BoM/CGBAPS staff at Cape Grim were/are largely responsible for the collection of archive samples and UEA flask air samples; the original (mid-1990s) subsampling of the archive for UEA was funded by AFEAS and CSIRO, ongoing subsampling by CSIRO. K. E. Adcock was supported by the UK Natural
Environment Research Council (PhD studentship NE/L002582/1). J. C. Laube received funding from the UK Natural Environment Research Council (Research Fellowship NE/I021918/1). Norfazrin Mohd Hanif has been funded through PhD studentship by the Ministry of Education Malaysia (MOE) and Universiti Kebangsaan Malaysia (UKM). We acknowledge use of the NAME atmospheric dispersion model and associated NWP meteorological data sets made available to us by the UK Met Office. We also acknowledge the significant storage
resources and analysis facilities made available to us on JASMIN by STFC CEDA along with the corresponding support teams.

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

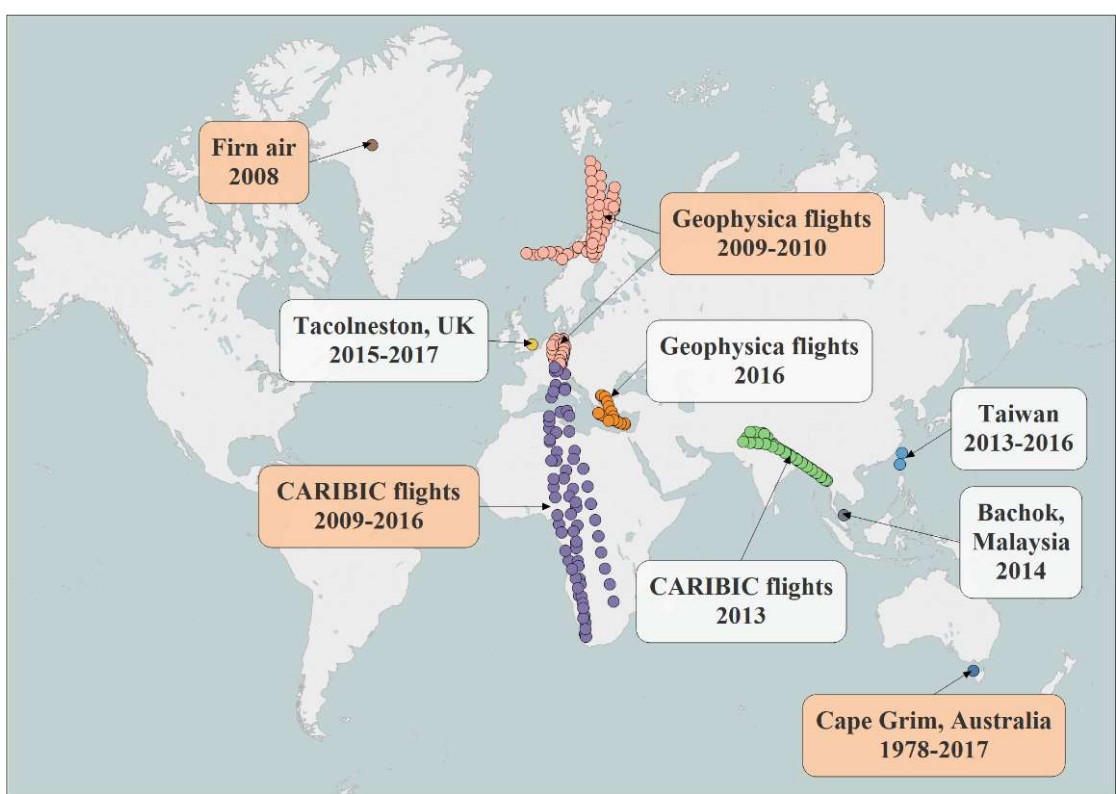

Figure 1. Sampling locations used in this study. Those locations that have been added since Laube et al. (2014) are in white. Those shaded orange featured in, or have been extended since, the Laube et al. paper.

Table 1. Air sampling campaigns from which atmospheric CFC-113a mixing ratios were measured, including the data published in Laube et al. (2014).

| Sampling campaign | Location | Longitude and Latitude | Dates | No. of samples | Nature of data |
|---|---|---|---|---|---|
| NEEM | Greenland | 77.445° N, 51.066° W 2484m a.s.l. | 14-Jul-2008– 30-Jul-2008 | 3 closest to the surface | Firn air surface data |

| | | | | | |
|---|---|---|---|---|---|
| Cape Grim | Tasmania, Australia | 40.683° S, 144.690° E | (07-Jul-1978) 14-Mar-2013– 23-Feb-2017 | 66 total, 20 new | Southern Hemisphere ground-based site |
| Taiwan | East Asia | Hengchun, 22.0547° N, 120.6995° E, (2013, 2015) Cape Fuguei, 25.297° N, 121.538° E, (2014, 2016) | 2013–2016 | 2013: 19 2014: 24 2015: 23 2016: 33 | Northern Hemisphere ground-based sites |
| Tacolneston Tower | Norfolk, United Kingdom | 52.3104° N, 1.0820° E | 13-Jul-2015– 16-Mar-2017 | 47 | Northern Hemisphere tall tower site |
| Bachok Marine Research Station | Bachok, Malaysia | 6.009° N, 102.425° E | 20-Jan-2014– 03-Feb-2014 | 16 | Tropical ground-based site |
| Geophysica flights 2009-2010 | North Sea | 76-48° N, 28-0° E | 30-Oct-2009– 02-Feb-2010 | 98 | Research aircraft |
| Geophysica flights 2016 | Mediterranean Sea | 33-41° N, 22-32° E | 01-Sep-2016 06-Sep-2016 | 23 | Research aircraft |
| CARIBIC flights | Germany to South Africa | 48° N-30° S, 6-19° E | 27-Oct-2009 28-Oct-2009 14-Nov-2010 20-Mar-2011 10-Feb-2015 13-Jan-2016 | 14 7 13 14 15 7 | Commercial aircraft |
| CARIBIC flights | Germany to Thailand | 32-17° N, 70-97° E | 21-Feb-2013 21-Mar-2013 09-Nov-2013 05-Dec-2013 | 14 7 14 14 | Commercial aircraft |

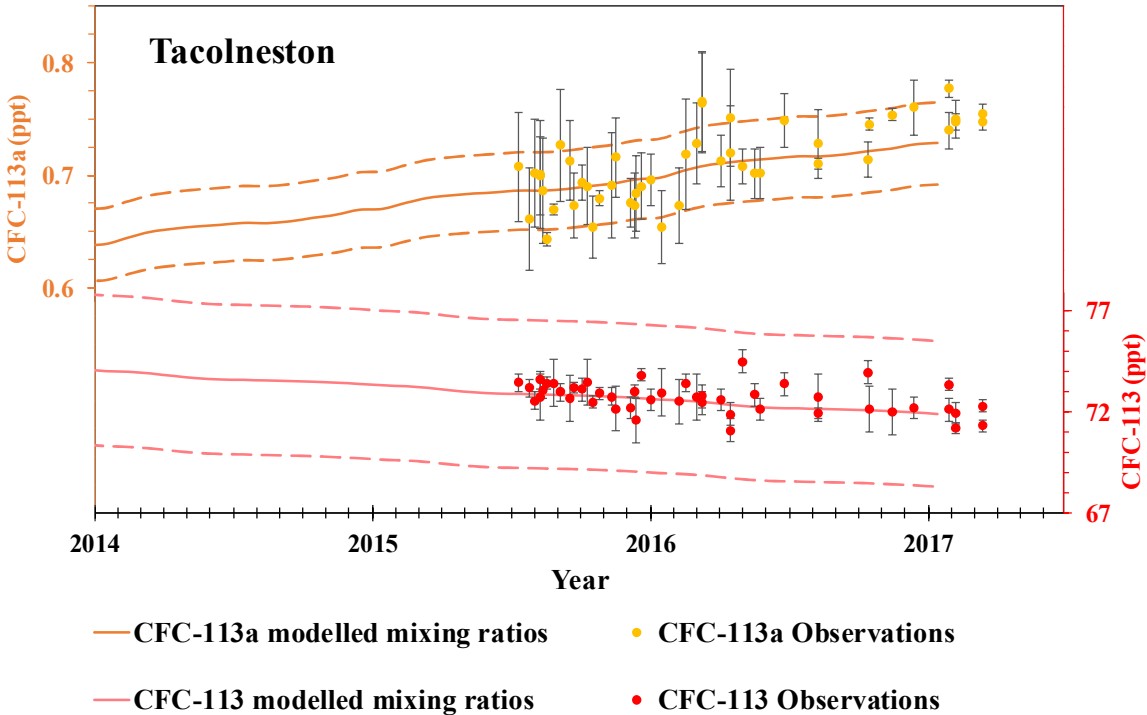

Figure 2. CFC-113a and CFC-113 modelled and observed mixing ratios at Tacolneston. The error bars represent the 1σ standard deviation. The modelled uncertainties are 5 % and are based on the model reproducing the reported mixing ratios of CFC-11 and CFC-12 at Cape Grim to within 5 % uncertainty (Reeves et al., 2005).

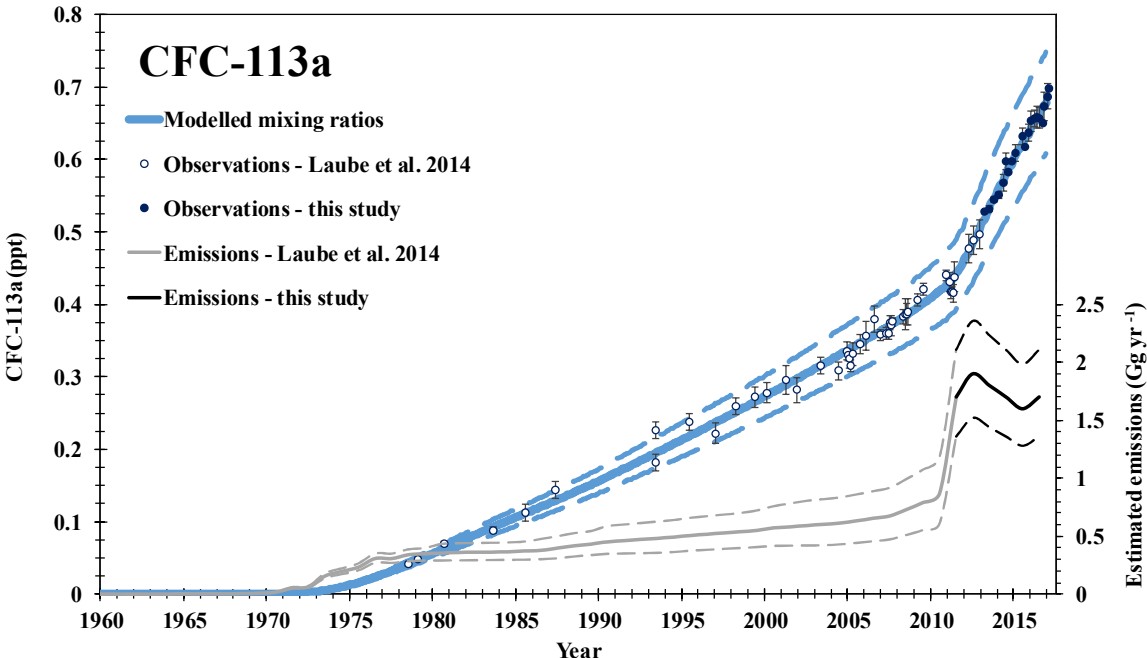

Figure 3. CFC-113a modelled and observed mixing ratios at Cape Grim 1960-2017 and estimated global annual emissions of CFC-113a. The observations are from July 1978-February 2017 with 1σ standard deviations as error bars. Data prior to 04-Dec-2012 is from Laube et al. (2014). The blue solid line represents the modelled mixing ratios with uncertainties (dashed blue line). The dashed black and grey lines represent the modelled 'best fit'

emissions with uncertainties (short-dashed). The method used for calculating the upper and lower emission bounds is in the supplementary material.

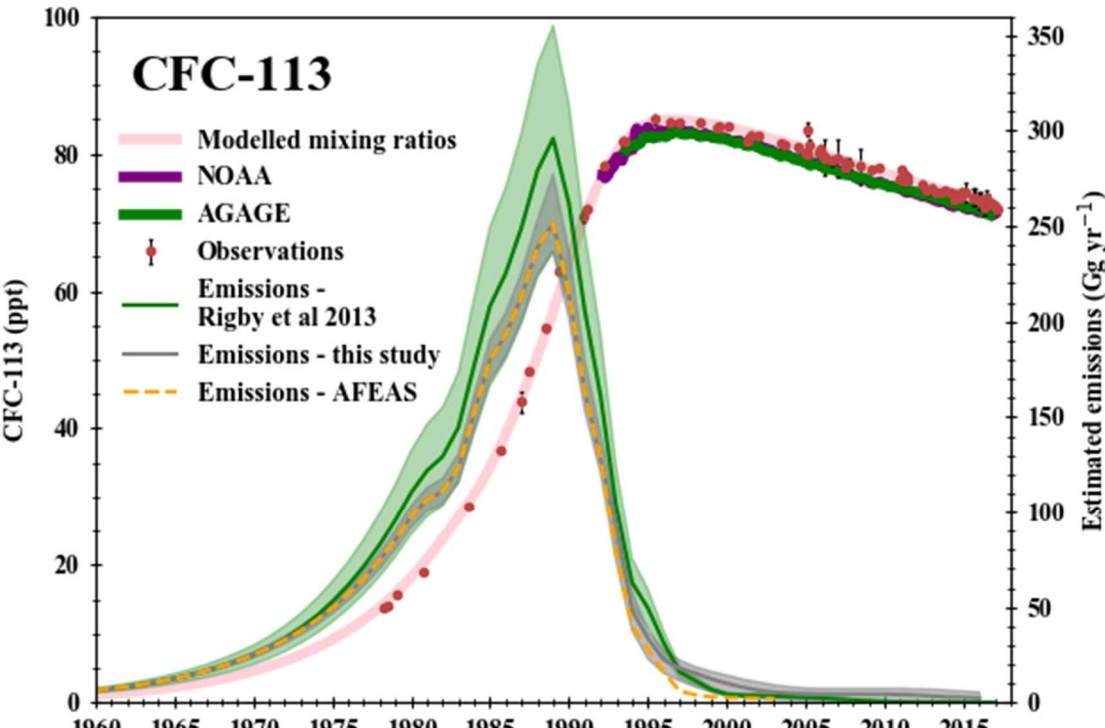

Figure 4. CFC-113 modelled and observed mixing ratios at Cape Grim 1960-2017 and estimated global annual emissions of CFC-113. The observations are from Cape Grim, Tasmania, July 1978-February 2017 with 1σ standard deviations as error bars.  Also for comparison are the NOAA and AGAGE CFC-113 mixing ratios at Cape Grim and previous emissions estimates from AFEAS and Rigby et al. (2013) (based on AGAGE in situ data) with 'likely' uncertainties (green lines).  The dashed black line shows the modelled 'best fit' emissions with uncertainties (grey lines). The method used for calculating the upper and lower emission bounds is in the supplementary material.

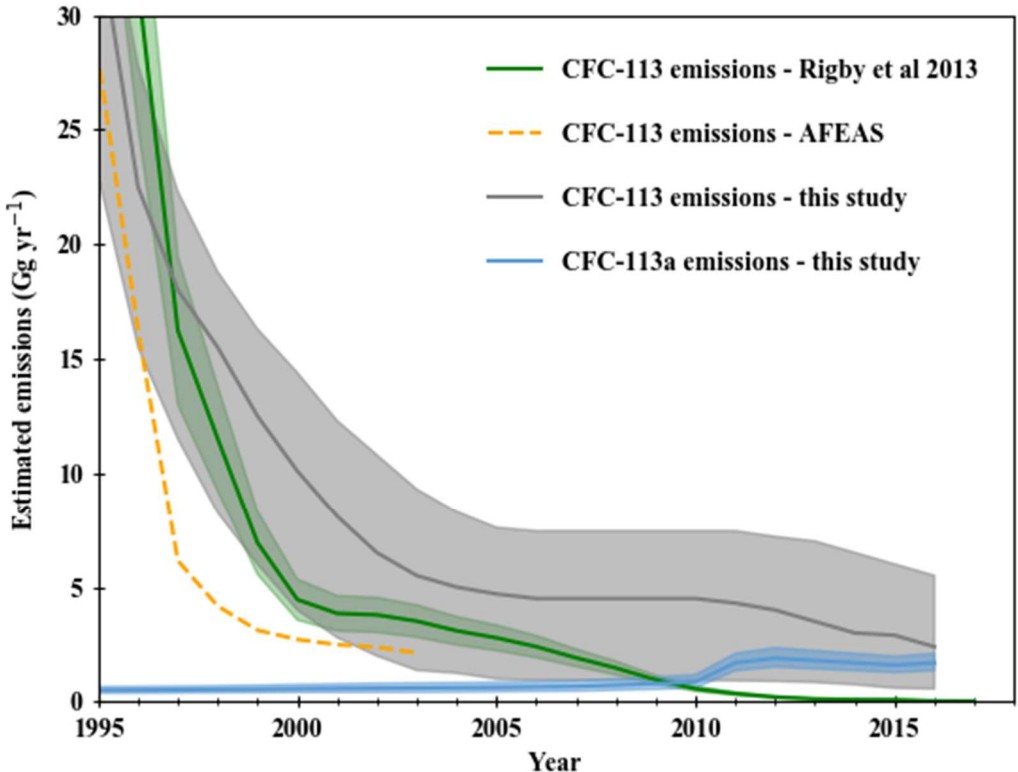

Figure 5. CFC-113 emissions from this study, AFEAS and Rigby et al 2013 and CFC-113a emissions from this study 1995-2016 with uncertainties.

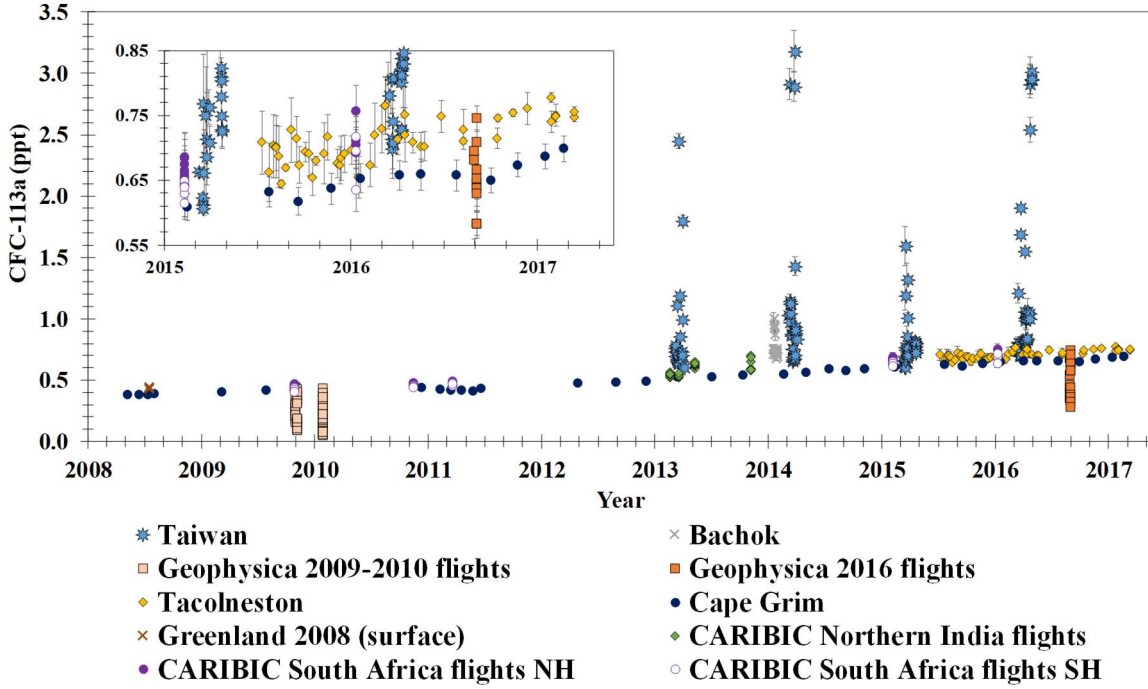

Figure 6. CFC-113a mixing ratios 2008-2017 from all the sources presented in this study with an inset of the period 2015-2017 to give an enlarged view of the Tacolneston data. The error bars represent the 1σ standard deviation.

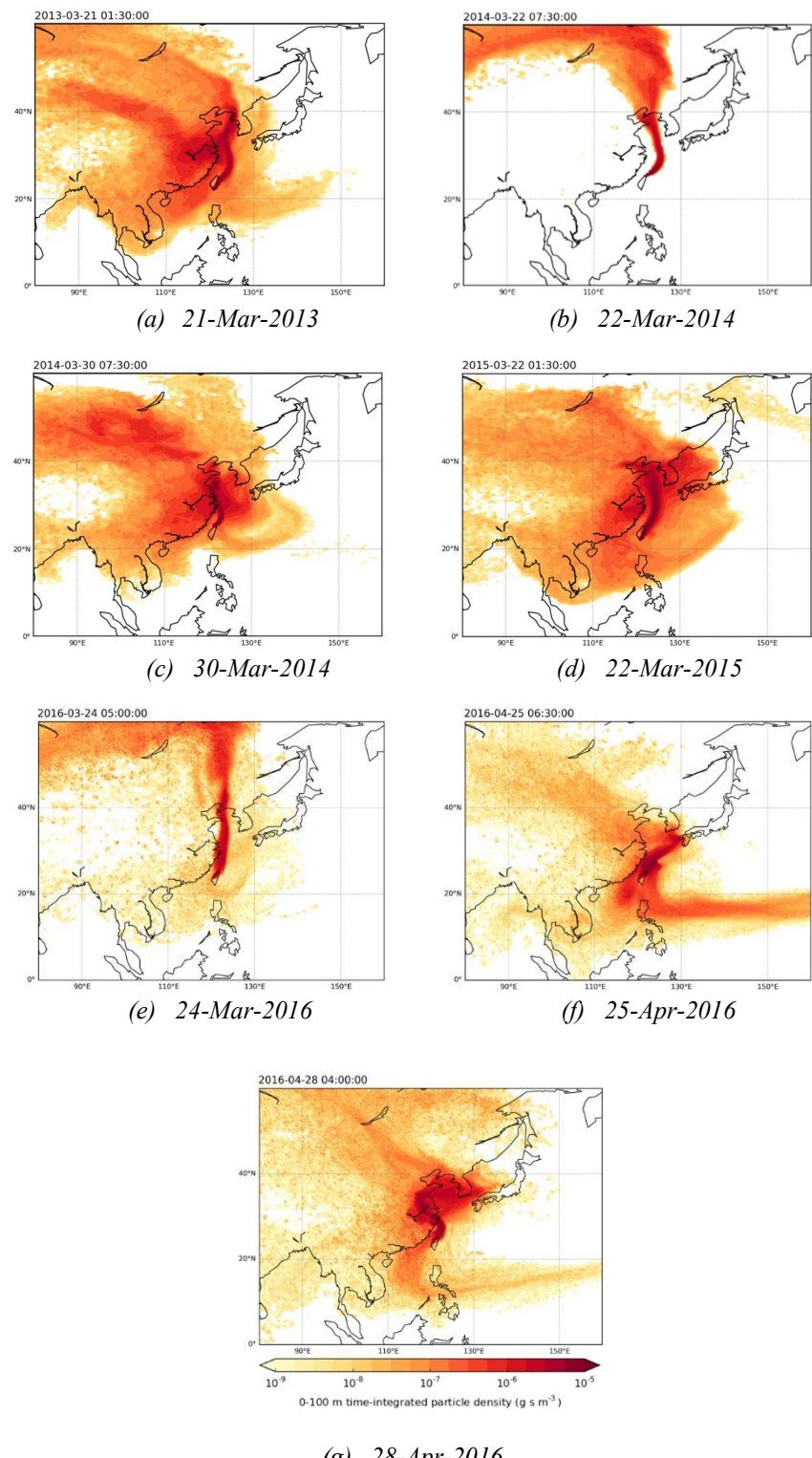

(a) 21-Mar-2013       (b) 22-Mar-2014

(c) 30-Mar-2014       (d) 22-Mar-2015

(e) 24-Mar-2016       (f) 25-Apr-2016

(g) 28-Apr-2016

Figure 7. NAME footprints derived from 12-day backward simulations and showing the time integrated density of particles below 100 m altitude for the approximate times when samples were collected during the Taiwan campaign. (a), (c), (d) and (g) are examples of one enhanced CFC-113a mixing ratio in each year. (f) is the sample taken just before (g) when the air was coming from a different direction and the mixing ratio of CFC-113a was much lower. (b) and (e) are also examples of samples with lower CFC-113a mixing ratios. Arrows in Fig. 8 show the mixing ratios of CFC-113a for these NAME footprints. For the rest of the NAME footprints see the supplementary material.

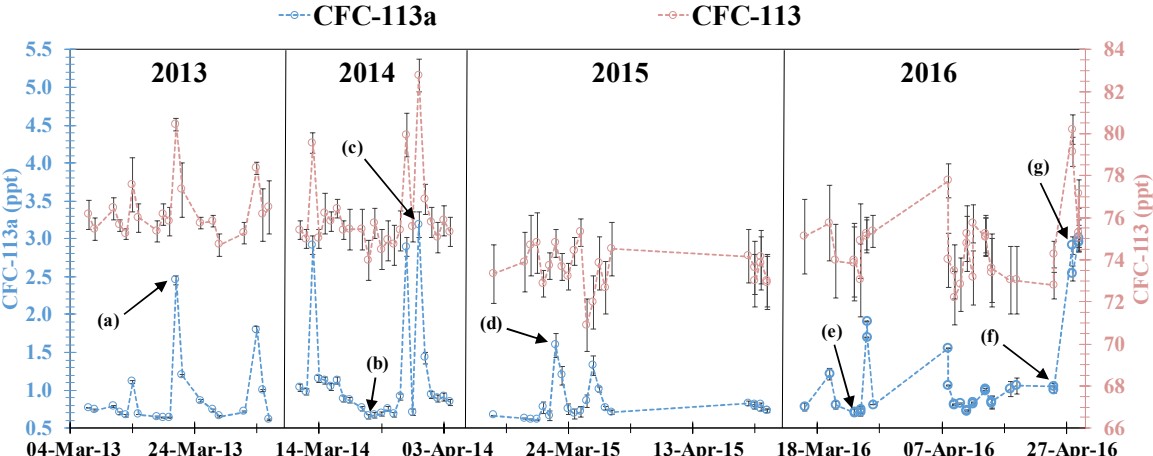

Figure 8. CFC-113a and CFC-113 mixing ratios observed in Taiwan in March and April 2013-2016. Arrows show the mixing ratios of CFC-113a that relate to the NAME footprints shown in Fig. 7. The error bars represent the 1σ standard deviation.

Table 2. Squared pearson correlations ($R^2$) of CFC-113a mixing ratios with other compounds in Taiwan 2013-2016.

|  | 2013 | 2014 | 2015 | 2016 |
|---|---|---|---|---|
| CFC-113 | 0.866 | 0.909 | 0.013 | 0.429 |
| HCFC-133a | 0.923 | 0.923 | 0.891 | 0.637 |
| HFC-134a | 0.001 | 0.055 | 0.010 | – |
| HFC-125 | 0.319 | 0.219 | 0.016 | 0.850 |
| CFC-114a | – | – | 0.754 | 0.386 |
| HCFC-123 | – | 0.013 | 0.217 | 0.202 |
| HCFC-124 | – | 0.537 | 0.833 | 0.078 |
| No. of data points | 19 | 24 | 23 | 33 |

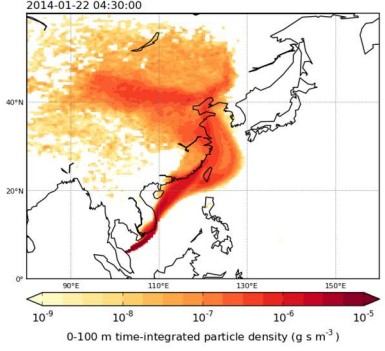

Figure 9. NAME footprint derived from 12-day backward simulation and showing the time integrated density of particles below 100 m altitude on 22-Jan-2014 during a period of elevated CFC-113a mixing ratios at Bachok, Malaysia.