# Peer review of "Continued increase of CFC-113a (CCl3CF3) mixing ratios in the global atmosphere: emissions, occurrence and potential sources"

_Atmospheric Chemistry and Physics, 2017_

## Referee Comment (RC1) · Anonymous Referee #1 · 15 Dec 2017

Review for Adcock et al., Continued increase of CFC-113a (CCl3CF3) mixing ratios in the global atmosphere: emissions, occurrence and potential sources

The authors provide updated data sets for CFC-113a from various observation platforms and use these to update global abundances and emissions. These show that after a rapid increase in CFC-113a emissions in approximately 2010, they have now leveled off. Interhemispheric gradients and pollution events at the Asian stations are presented. Potential sources of CFC-113a are discussed, which however remain speculative.

The paper is an informative and useful update on a compound, little is known about.

[Figure]

This study adds to other recent studies of newly-discovered compounds with no purposeful end use, and shows the necessity to distinguish the two isomers in order to de-tangle their stories. The paper is well written and understandable, but could benefit from some consolidating of information, in particular in the sections where the potential sources are discussed. The topic is well researched and the authors have done a good job in bringing the many data sets together. The supplement adds more information and data are made available. This paper is well suited for publication in ACP and I have only minor comments, which, once thoroughly handled, should make this acceptable for publication.

Introduction: A brief discussion on the loss mechanisms of CFC-113a would be helpful to understand the cycling of this compound through the atmosphere, in particular for the discussion on CFC-113a in the stratosphere.

p. 2, l. 77: What kind of pumps were used for the various sampling sites?

p. 2, l. 84: Please elaborate more on the use of different methods (chromatography columns) particularly to give insight which samples / batches of samples were measured one way or the other.

p. 2, l. 86: Primary calibration scale: Describe, or refer to the literature, on how the primary reference material was produced (how many primary standards, at what ppt level), and what the estimated accuracy is. Was the CFC-113a pure or contaminated with CFC-113? Accuracy of the NOAA calibration scale for CFC-113.

p. 2, l. 89: How did the repeated standard measurements feed into the calculation? Were these referenced against other standards, or were chromatographic peak sizes assumed constant over a day? How many standards were measured per day?

p. 2, Methods. If not published previously, please provide a few more analytical details. How well do CFC-113 and CFC-113a separate on the two columns used, which fragments were measured? If available, provide a spectrum of CFC-113a; how well

does it compare with one from the literature?. If some ions were used for both isomers, what were their sensitivities if scaled to the same mole fraction on the MS used for this study? Does 'pure' CFC-113 normally have CFC-113a impurities or vice versa? Provide any other information that can be useful for comparison with other networks, which may not be able to separate the two isomers (this could all be in the supplement).

p. 3, l. 95: How many CGAA samples were added, were these equally spaces over the 2012 – 2017 period. Were some of the earlier samples from Laube et al., 2014 re-analyzed and if so, how did these agree with the present study?

p. 3, l. 109: Which is day and which is month in the xx/zz/yyyy date description. This can be rather confusing (and seems to be reversed in the supplementary tables), why not spell out the month(s), like e.g. p. 7 l. 290? This confusing style re-appears throughout the document.

p. 3, l. 124: The very first mentioning of CFC-113 here without any prior motivation for this leaves the reader confused.

p. 3, l. 125: Even though explained in details in the cited references, please provide a couple of sentences describing this model. Are the emissions derived from an inversion, was a prior used?

p. 3, l. 127: What is the UEA Air Archive?

p. 3, l. 126: Have the CFC-113 mixing ratios been published previously, or are these measurements part of the present study? By reading this sentence it is not a priori clear if AFEAS has also provided bottom-up emissions for CFC-113a.

p. 3, l. 131: What do the authors mean by 'isolated'?

p. 3, l. 138: Please explain why Tacolneston can be considered of NH background air given that there is apparently a gradient of CFC-113a in the NH. The absence of pollution events at Tacolneston may make this site representative for background air at that latitude at most. See similar issue p. 6 l. 268. 'Spikes' is a rather confusing term.

p. 4, l. 141. I am not convinced about the shift to more southerly latitudes. Is this an assumption that was made, or a result of the model? If the former, on what basis is this assumption founded? If the latter, how can a model using Cape Grim data give such detailed results about the NH distribution, particularly when apparently Cape Grim is insensitive to the NH distribution as stated on p. 3, l. 133? The mentioning of Tacolneston observations here is confusing also, how could these be used to determine the NH latitudinal distribution over decades if there are only a few years of measurements? Also the comparison to Taiwan seems inappropriate if the authors use median values for that site rather than a 'Taiwan background'. Overall there seem to be too many degrees of freedom here to be able to pin-point such a detailed evolution of NH latitudinal gradients.

p. 4, l. 171 (Overall . . .) Is this sentence necessary?

p. 4, l. 176: Is the 0.03 ppt/yr increase a result of the model or some other kind of fit? The mentioning of 0.06 ppt/yr for Taiwan is somewhat missleading as it makes the reader believe that mixing ratios have grown more rapidly there, but isn't the Taiwan growth most likely tagged with a large uncertainty given the fact that the authors take the median values rather than deriving a real 'background' record for Taiwan.

p. 4 l. 182: The presentation of the result would evolve less confusing if the emissions were first mentioned before rates of growth of these emissions are discussed. Also, the mentioning of an average of 1.7 Gg/yr does not say anything about the variability of the emissions in these years (There is not a single mentioning of emissions for a specific year). It would be helpful to learn how much they varied in these years, by e.g. giving the range of emission for these few years. I presume that the numbers in parentheses are some uncertainty ranges, or are these the ranges they vary over that time?

p. 5, l. 185: Can this 2% offset (using the same primary calibration scale) be related that perhaps NOAA measures the combined CFC-113a/CFC-113 isomers? Also, there does not seem to be a description anywhere on how the NOAA scale for CFC-113 was

adopted, and what the propagation uncertainties of this procedures might be. Which way is the offset, which network reports higher values?

p. 5, l. 200: It would be very informative to see a graphical comparison of the emissions for the two compounds over the last years, for example by adding those of CFC-113 to Fig 3, or to somewhere plot the ratio of the CFC-113a/CFC-113 emissions.

p. 5, l. 211: This is confusing, if the AFEAS data are used as the CFC-113 emissions in the model, then the sentence that follows does not make sense. Do the authors perhaps mean that the modeled emissions agree closely with the AFEAS bottom-up emissions?

Fig. 4: colors are very hard to distinguish, for example, colors for NOAA and modelled mixing ratios seem the same. Also, filled circles don't appear as such due to their close proximities. Dashed lines appear as dash-dotted in the legend. The matching of modeled and AFEAS data is perplexing – are AFEAS data used as prior? In which year does AFEAS end, 2000; it is not possible to see this on the graph? I see only one green Rigby uncertainty line. Are the uncertainties shown for the present study based on the 'likely' or 'possible' range (p. 5 l. 22) or something else?.

p. 5, l. 221: Lifetime work was really done by SPARC, it might be better to cite that work.

p. 5, l. 234: Do you mean 'related to CFC-113 emissions from (old) banks, i.e impurities of CFC-113a in CFC-113? This would not agree with their historic emission ratios. Please clarify.

p. 6, l. 243: I suggest to phrase the other way: When there are spikes, then the NAME model . . ..

p. 6, l. 252: Were compounds other than those listed in Table 2 also looked at?

p. 6, l. 277: 'generally': Does this mean that this was not always the case and that some Cape Grim results were higher than Tacolneston. From Fig 5 inset, it is not easy

to see this. Is the dark filled circle at the beginning of 2016 at ∼0.75 ppt from Cape Grim (with uncertainties smaller than the plotting symbol)? l. 279: replace 'higher' by 'lower'.

p. 7, l 289: It is unclear, if only the three surface samples were used here for the inter-hemispheric comparison, or a historic reconstruction of the NH based on more samples. If only the surface samples were use, the mentioning of 'trend' is not adequate. Also, Greenland firn air samples are here declared as 'representative of background Northern Hemisphere CFC-113a mixing ratio', which (similar to the same statement for Tacolneston) seems to be in contradiction to the assumption of latitudinal gradients in the NH. The samples are hard to see in Fig. 5, perhaps but a year in the legend for 'Greenland (surface)'.

p. 7, l. 319: What is the 's' in parentheses?

p. 8, l. 334: I disagree with the statement on l. 333 ff, that the absence of a correlation is not what we would expect. Before, the authors rightly state, that the correlation between CFC-113 and CFC-113a may derive from co-located factories (this is a rather likely scenario, as there are large centers of industrial activities in China). Taiwan may simply see air from places where CFC-113 and CFC-113a factories are not co-located,

p. 8, l. 347: I am having difficulties to derive from Manzer 1990, that the CFC-113a route is a main pathway. It appears that he showed many pathways graphically, and listed two examples of potential pathways, neither of which was via CFC-113a. In contrast, Maranion et al., seem to suggest this route, but without backing it up with literature.

References: Laube et al., 2014: Captial 'A' For Brenninkmeijer, C. A. M.; Manzer, 1990. Reverse intials; Rigby et al., 2013: Correct Muhle

Fig. 2: Limiting the y axes labels to the range of the data would greatly help in distinguishing the two data sets. Are these error bars now also including some uncertainties

of the standard measurements, as indicated in the main text?

Fig. 3: Similar comment as for Fig 2 for the right y-axis: Label ticks only in range where data are. For the mean emissions, a solid line rather than a dashed line would help to see the apex better. Date 04/12/2012 not apriori clear, which is month and which is day. Legend shows light blue filled circles but these can't be seen in the plot. Suggest to replace these by lines for the model results.

Consider to somehow show the CFC-113 emissions for the last decades in a different form, such that they could be compared to the CFC-113a emissions in Fig. 3 (perhaps add them there).

Fig. 5: It is very difficult to distinguish the various sampling sets. Please improve figure.

Fig. 7: There seem to be no error bars on the CFC-113a measurements.

Fig. 8: Why was this particular measurement chosen? There seem to be many elevated CFC-113a shown in Fig. 5.

SI Tables: Emissions are calculated for 2017 based on 2 samples only for Jan and Feb 2017. This is almost certainly biased and I suggest to omit presenting model results for 2017.

Consider complementing tables with informations on calibration scales used for the various compounds and data sets.

---

## Referee Comment (RC2) · Anonymous Referee #2 · 21 Dec 2017

This paper updates and advances our understanding of CFC-113a in the global atmosphere, its lifetime, regions contributing emissions, and potential sources. The continued increase is interesting and important to document and understand, especially given the accelerated increase that appears to be continuing after being initially documented in an earlier paper. It is nice to see the broad range of measurement locations and information they supply. The paper is mostly sound, although there are a few sections where some reconsideration of results is warranted and where some improvement in the writing is needed. But after these issues are addressed, the paper likely would be appropriate for publication.

[Figure]

Issues to consider: As the authors note, the impact of these emissions on the ozone layer to date is minor. Suggesting that more CFC-113a might make it to the stratosphere than is indicated by surface means is a conclusion whose importance can only be speculated about (line 309-312). It is not a conclusion based on data presented here so doesn't seem appropriate to include. Geophysica results from the stratosphere are indicated as starting at background levels and decreasing above; in other words, entirely consistent with background mole fractions at Earth's surface.

Related to this, it seems important to mention in the text that a constant emission of 2 Gg/yr for a chemical with a 50-yr lifetime yields a steady-state global mole fraction of 5 ppt (15 ppt of Cl for CFC-113a). This helps the reader to objectively understand the significance of these results compared to the contribution of other chemicals including CFCs in a much more meaningful way than a comparison of cumulative emissions since 2007, for example (lines 206-209). Undoubtedly CFC-113a emissions could increase, but the potential for this is constrained by the cause of the increasing emissions and, for that fraction associated with HFC production, the Kigali Amendment.

Potential sources for CFC-113a emissions should be considered in light of the fact that emissions were fairly small until 2010, and then increased to a new value and have been essentially constant since. It is my view that this step change in emissions is primarily why this paper is worth publishing in ACP. It provides a strong hint as to which process likely caused this step change (at the least it reduces the likelihood of some causes) and is important to consider in gauging the likelihood of emissions increasing in the future. At the present time this section (4) rambles a bit and would benefit from significant tightening.

On the upper lifetime limit derived for CFC-113 based on the observed rate of change of CFC-113. Some consideration or discussion of steady-state lifetimes vs lifetimes at zero emissions is required here before such a conclusion is made. These are two different quantities that have different values. For CFC-113, I expect its lifetime in the presence of zero emissions to be slightly shorter than one derived at SS. See papers by

Prather on this topic, and consider calculating the difference in your model to determine if an upper limit to a SS lifetime is inconsistent or not with the observationally-derived value upper limit (assuming E=0).

Line 438-440. A suggestion that new regulatory mechanisms might need to be added to the Montreal Protocol is made in the conclusion. This statement diminishes the objective nature of the data and discussion included in the paper. Policy is made with consideration of a broad range of costs and benefits, and you cannot begin to cover this complex and multifaceted discussion in a paper about atmospheric changes. If you want to comment on policy, consider doing it with an "if...then" construction. And in this case, if policy-makers wanted to require absolutely zero emissions of CFCs, then they might consider doing x,y, and z.

The discussion of section 4 in the supplement is not useful without explicitly considering the changes over time in tropospheric entry values. Without this, the section adds little to the paper.

Other items. How was calibration consistency maintained throughout time and across the different missions?

Figure 3, mention blue solid and dashed lines in caption.

p. 2, line 89-91, uncertainties are mentioned, but these are not the uncertainties used in the modeling, which are discussed in the supplement but not the section on modeling. I'd suggest that this appear somewhere in the main text.

On uncertainties in CFC-113 calibration arising from co-elution of CFC-113a. Consider doing the atmospheric measurement community a favor by discussing the relative magnitude of interference that an analyst might have in measuring CFC-113 given co-elution of CFC-113a at the different ions these chemicals have in common (perhaps a simple table in the supplement?). This would be very helpful, and easy to add, I imagine, given that you are in a unique position to supply this important information

that to first order would be independent of mass spec instrument being used.

Lines 133-136 and 150-152. This doesn't make sense. Fitting well data at CGO wouldn't say much about the accuracy of and emission distribution in the lower SH and throughout the NH.

Lines 217, Confusing phrasing. Ultimately, global emission magnitudes derived from observations depend on the lifetime used, and you used different lifetimes than others.

Paragraph starting on line 251. First part: make this a discussion of variability in mole fractions and not just mole fractions. This makes your point valid and will help later when you are discussing differences in trends vs short-term variability in the UK vs Cape Grim. Second point: mention the HCFC-133a lifetime.

Para starting on line 265 (also line 291). Assertions are made that are not valid here or that extend limited results to broader context without justification (was Tacolneston sensitive to emissions from all UK source regions? Why would results from this site be representative of the NH? They might be proportional to that quantity, but not necessarily quantitatively the same). These are weakness to the paper that aren't needed and could be easily avoided.

---

## Author Response (AR1)

[revised manuscript text omitted]

The authors provide updated data sets for CFC-113a from various observation platforms and use these to update global abundances and emissions. These show that after a rapid increase in CFC-113a emissions in approximately 2010, they have now leveled off. Interhemispheric gradients and pollution events at the Asian stations are presented. Potential sources of CFC-113a are discussed, which however remain speculative.

The paper is an informative and useful update on a compound, little is known about. This study adds to other recent studies of newly-discovered compounds with no purposeful end use, and shows the necessity to distinguish the two isomers in order to de-tangle their stories.

The paper is well written and understandable, but could benefit from some consolidating of information, in particular in the sections where the potential sources are discussed.

The topic is well researched and the authors have done a good job in bringing the many data sets together. The supplement adds more information and data are made available. This paper is well suited for publication in ACP and I have only minor comments, which, once thoroughly handled, should make this acceptable for publication.

1. Introduction: A brief discussion on the loss mechanisms of CFC-113a would be helpful to understand the cycling of this compound through the atmosphere, in particular for the discussion on CFC-113a in the stratosphere.

We agree with the reviewer that a brief discussion on the loss mechanisms would be helpful and we have added a sentence explaining the loss mechanisms of CFCs in general to the introduction (lines 47-49). We thought it would fit better in the introduction if we talked about the loss mechanisms of CFCs in general rather than CFC-113a specifically.

"CFCs have negligible loss mechanisms in the troposphere and only break down when they reach the stratosphere where they are exposed to strong ultraviolet light and decompose mostly through photolysis and reaction with $O^1D$ (Ko et al., 2013)."

2. p. 2, l. 77: What kind of pumps were used for the various sampling sites?

We agree with the reviewer that information of the kinds of pumps used would be a good idea. Various pumps were used for the different sampling activities and they have been described in previous articles: Cape Grim (Allin et al., ACP, 2015); Taiwan, Tacolneston and Bachok (Oram et al., 2017); Geophysica (Laube et al., ACP, 2010 (Fractional Release…)); CARIBIC (Brenninkmeijer et al., 2007).

This sentence was added to the methods section (lines 84-86)

"Various pumps were used for the different sampling activities, all of which have been thoroughly tested for a large range of trace gases (e.g. Brenninkmeijer et al., 2007; Laube et al., 2010; Allin et al., 2015 and Oram et al., 2017)."

3. p. 2, l. 84: Please elaborate more on the use of different methods (chromatography columns) particularly to give insight which samples / batches of samples were measured one way or the other.

We agree with the reviewer that more information of the different columns should be added. We have added the Cape Grim CFC-113a and CFC-113 measurements on the GS GasPro column (length ~50 m, ID 0.32 mm) and the KCl-passivated $Al_2O_3$-PLOT column (length: 50 m, ID 0.32 mm) to the spreadsheet in the supplementary material. A range of Cape Grim samples were reanalysed on the AlPLOT column and showed very good agreement with the previous GasPro column-based measurement with comparable precisions and no apparent offset. We have also added two tables to the supplement spreadsheet: one for the Taiwan measurements and one for all the other samples showing which columns were used for which measurements.

Some of the samples collected in Taiwan in 2013 were also measured on another GC-MS. HCFC-133a, HFC-134a & HFC-125 were measured on the Entech GCMS. Some information about this instrument was added to the methods section (lines 101-103).

"The samples collected in Taiwan in 2013 were also measured on the Entech-Agilent GC-MS system operating in electron ionisation (EI) mode. This consists of a preconcentration unit (Entech model 7100) connected to an Agilent 6890 GC and 5973 quadrupole MS (Leedham Elvidge et al., 2015)."

4. p. 2, l. 86: Primary calibration scale: Describe, or refer to the literature, on how the primary reference material was produced (how many primary standards, at what ppt level), and what

the estimated accuracy is. Was the CFC-113a pure or contaminated with CFC-113? Accuracy of the NOAA calibration scale for CFC-113.

We believe all of this information is already available and it is not necessary to add it to the paper. For CFC-113a all of the requested information was published in Laube et al., 2014 who reported the first measurements of this compound in the atmosphere. For CFC-113, which we are reporting on a NOAA scale, all of the information is publicly available from the NOAA ESRL websites. NOAA calibration scales are internationally widely recognised and Laube et al., 2013 demonstrated that UEA measurements of CFC-113 on samples collected at Cape Grim match respective NOAA observations of that gas quite closely and over several decades with a small offset.

5.  p. 2, l. 89: How did the repeated standard measurements feed into the calculation? Were these referenced against other standards, or were chromatographic peak sizes assumed constant over a day? How many standards were measured per day?

We have added these sentences to the methods section (lines 105-107)

"On a typical day, the working standard is measured five to eight times, between every two or three samples. The sample peak sizes are measured relative to the standards measured just before and after them. The working standard is used to correct for small changes in instrument response over the course of a day."

As described in the manuscript (lines 109-111), the repeated standard measurements feed into the calculation as follows: "The measurement uncertainties are calculated the same way for all measurements and represent one sigma standard deviation. They are based on the square root of the sum of the squared uncertainties from sample repeats and repeated measurements of an air standard on the same day."

6.  p. 2, Methods. If not published previously, please provide a few more analytical details. How well do CFC-113 and CFC-113a separate on the two columns used, which fragments were measured? If available, provide a spectrum of CFC-113a; how well does it compare with one from the literature? If some ions were used for both isomers, what were their sensitivities if scaled to the same mole fraction on the MS used for this study? Does 'pure' CFC-113 normally have CFC-113a impurities or vice versa? Provide any other information that can be useful for comparison with other networks, which may not be able to separate the two isomers (this could all be in the supplement).

Almost all available information on GasPro column-based measurements has already been published in Laube et al., 2014. We only add here that a possible interference could arise when measuring CFC-113a on that column using m/z 116.91 if concentrations of the nearby eluding HCFC-123 are high. This was the case for a small number of samples analysed for this work and those measurements were either a) repeated using the interference-free m/z 120.90, b) replaced with measurements on the other column, or c) excluded. The KCl-passivated $Al_2O_3$-PLOT column separated CFC-113 and CFC-113a well, we observed no interferences and used m/z 116.91 for quantification. We have added this information to the methods section (lines 94-99). Whether pure CFC-113 normally contains CFC-113a impurities would require diluting and analysing multiple samples from multiple companies which exceeds the scope of this study. We would however be very open to carry out a direct intercomparison experiment if approached.

950   7. p. 3, l. 95: How many CGAA samples were added, were these equally spaces over the 2012 –
      2017 period. Were some of the earlier samples from Laube et al., 2014 re-analyzed and if so,
      how did these agree with the present study?

      We added this sentence to the methods section (line 116-117)
955

      "From 2013 to 2017, 20 samples were collected at Cape Grim at irregular intervals of between one
      to five months apart."

      Some of the earlier samples from Laube et al 2013 and Laube et al 2014 were re-analysed on the
960   KCl-passivated $Al_2O_3$-PLOT column (length: 50 m, ID 0.32 mm). They showed very good agreement
      with the previous GasPro column-based measurement with comparable precisions and no apparent
      offset. We have added this information to the methods section (lines 122-124). We have added the
      Cape Grim CFC-113a and CFC-113 measurements on the GS GasPro column (length ~50 m, ID 0.32
      mm) and the KCl-passivated $Al_2O_3$-PLOT column (length: 50 m, ID 0.32 mm) to the spreadsheet in the
965   supplementary material.

      8. p. 3, l. 109: Which is day and which is month in the xx/zz/yyyy date description. This can be
      rather confusing (and seems to be reversed in the supplementary tables), why not spell out the
      month(s), like e.g. p. 7 l. 290? This confusing style re-appears throughout the document.
970

      We agree with the reviewer and all the dates in the article and in the supplementary material have
      been changed to the format dd-mmm-yyyy.  We have not changed the date format in the NAME
      model figures but have added a sentence to the methods section (lines 208-209) that says "Dates in
      the NAME footprint maps are presented in the format yyyy-mm-dd and use UTC time."
975

      9. p. 3, l. 124: The very first mentioning of CFC-113 here without any prior motivation for this
      leaves the reader confused.

      We refer the reviewer to the abstract, which states that "We compare the long-term trends and
980   emissions of CFC-113a to those of its structural isomer, CFC-113 ($CClF_2CCl_2F$), which still has much
      higher mixing ratios than CFC-113a, despite its mixing ratios and emissions decreasing since the
      1990s."

      We also modified the first sentence in this section (lines 150-151) to:
985

      "A two-dimensional atmospheric chemistry-transport model was used to estimate, top-down, global
      annual emissions of CFC-113a and CFC-113 for the purpose of comparing the emissions of the two
      isomers."

990   10. p. 3, l. 125: Even though explained in details in the cited references, please provide a couple of
      sentences describing this model. Are the emissions derived from an inversion, was a prior used?

      For this comment we mostly moved information from the supplementary material into the paper to
      give a more detailed description of the model (lines 151-156).
995

      "The model contains 12 horizontal layers each representing 2 km of the atmosphere and 24 equal-
      area zonally averaged latitudinal bands. The modelled mixing ratios for the latitude band that Cape
      Grim is located within (35.7° S–41.8° S) were matched as closely as possible to the observations at
      Cape Grim (40.7° S) by iteratively adjusting the global emissions rate until the differences between

1000 the modelled mixing ratios and the observations were minimised. For more details about the model see Newland et al. (2013); and Laube et al. (2016)."

11. p. 3, l. 127: What is the UEA Air Archive?

1005 We have changed the sentence (lines 117-119):

"The CFC-113 mixing ratios (1978-2017) from analyses of archived air samples collected at Cape Grim, Tasmania and analysed at the UEA, together with NOAA flask data, and AGAGE *in situ* data are also included to compare the two isomers."

1010
In addition, all later references to the "UEA Air Archive" have been changed to the "UEA Cape Grim data set".

12. p. 3, l. 126: Have the CFC-113 mixing ratios been published previously, or are these
1015 measurements part of the present study?

Most of the UEA Cape Grim CFC-113 mixing ratios were published in Laube et al., 2013, which also included the aforementioned comparison to NOAA measurements from the same site.

1020 We have added this sentence to the methods section (lines 121-122)

"Most of the CFC-113 UEA Cape Grim data set was previously published in (Laube et al., 2013)."

13. By reading this sentence it is not a priori clear if AFEAS has also provided bottom-up emissions
1025 for CFC-113a.

AFEAS has not provided bottom-up emissions for CFC-113a. That part of the sentence refers only to CFC-113. We have now separated it into two sentences (lines 157-162).

1030 "We now update the CFC-113a emission estimates using an additional four years of Cape Grim measurements. The CFC-113 emissions are estimated using CFC-113 mixing ratios at Cape Grim for 1978-2017 from the UEA Cape Grim dataset and compared with bottom-up emissions estimates from the Alternative Fluorocarbons Environmental Acceptability Study (AFEAS, https://agage.mit.edu/data/afeas-data)."

1035
14. p. 3, l. 131: What do the authors mean by 'isolated'?

We agree with reviewer and have replaced the word in question with the more appropriate "remote" (line 169).

1040
15. p. 3, l. 138: Please explain why Tacolneston can be considered of NH background air given that there is apparently a gradient of CFC-113a in the NH. The absence of pollution events at Tacolneston may make this site representative for background air at that latitude at most. See similar issue p. 6 l. 268. 'Spikes' is a rather confusing term.

1045
We agree with the reviewer and have changed this sentence (lines 177-178):

"Tacolneston can be considered to be representative of Northern Hemisphere background mixing ratios of CFC-113a **for that latitude** as there are no **significant enhancements** in mixing ratios (Figure 1050 2)."

In addition, all later references to "spikes" have been changed to "enhancements".

16. p. 4, l. 141. I am not convinced about the shift to more southerly latitudes. Is this an assumption that was made, or a result of the model? If the former, on what basis is this assumption founded? If the latter, how can a model using Cape Grim data give such detailed results about the NH distribution, particularly when apparently Cape Grim is insensitive to the NH distribution as stated on p. 3, l. 133? The mentioning of Tacolneston observations here is confusing also, how could these be used to determine the NH latitudinal distribution over decades if there are only a few years of measurements? Also the comparison to Taiwan seems inappropriate if the authors use median values for that site rather than a 'Taiwan background'. Overall there seem to be too many degrees of freedom here to be able to pin-point such a detailed evolution of NH latitudinal gradients.

We agree with the reviewer that there are many degrees of freedom here, which is why we state at the end of the paragraph that "It should be noted that while there is evidence that supports the emission distribution used here, there might be alternative distributions that result in equally good fits to the trends, particularly in the earlier part of the record." (lines 192-194). We also note, however, that for the later part of the trend (which is the main focus of this manuscript) the assumed emission distribution gives the best match to NH mixing ratios both in Tacolneston and in Taiwan. We would also like to point out that medians are mentioned nowhere in this paragraph, when we mention "measurements from Taiwan" we mean that there are significant enhancements in CFC-113a mixing ratios in Taiwan indicating continued emissions so it is not unreasonable to move emissions to this latitude band in the model. We have modified this sentence (lines 184-186):

"There are significant enhancements in CFC-113a mixing ratios in our measurements from Taiwan indicating continued emissions in this region (Section 3.2.1) which is consistent with emissions in this latitude band in the model."

17. p. 4, l. 171 (Overall : : :) Is this sentence necessary?

We agree with the reviewer that this sentence is not necessary and we have deleted this sentence and modified the next sentence (lines 216-218):

"Between 1978 and 2009 the average rate of increase was 0.012 ppt yr$^{-1}$; then between 2010 and 2017 the rate had risen 3-fold to about 0.037 ppt yr$^{-1}$."

18. p. 4, l. 176: Is the 0.03 ppt/yr increase a result of the model or some other kind of fit?

The first measurement at Tacolneston was 0.71 ppt on 13-Jul-15 at 11:03 and the most recent measurement for which we have a CFC-113a mixing ratio is 0.76 ppt on 16-Mar-17 at 16:30. The calculation is the difference between these two values (0.76-0.71=0.05ppt) divided by 20 (because there are 20 months between these two measurements) to give an increase per month. Then this is multiplied by 12 to give an average increase per year of 0.03 ppt.

We have changed the sentence (lines 219-221):

"Although measurements at Tacolneston were made for a shorter time period (20 months), it also experienced an increase in CFC-113a mixing ratios of 0.03 ppt yr$^{-1}$ over the period July 2015 to March 2017 **based on start and end points** (Figure 2)."

19. The mentioning of 0.06 ppt/yr for Taiwan is somewhat missleading as it makes the reader believe that mixing ratios have grown more rapidly there, but isn't the Taiwan growth most likely tagged with a large uncertainty given the fact that the authors take the median values rather than deriving a real 'background' record for Taiwan.

We agree with the reviewer, we do not have enough data to reliably derive background trends. We have deleted the sentence about Taiwan. We has also deleted the sentence explaining the median average in the methods section (lines 146-148) because it is the only place the Taiwan median is used.

"the median mixing ratios of CFC-113a in Taiwan increased on average by 0.06 ppt yr$^{-1}$ from 2013 to 2016."

"The median mixing ratios are used for the measurements made at the Taiwan sites to decrease the influence of the large spikes in CFC-113a mixing ratios that occurred during these campaigns (Section 3.2.1). All other averages are calculated using the mean."

20. p. 4 l. 182: The presentation of the result would evolve less confusing if the emissions were first mentioned before rates of growth of these emissions are discussed. Also, the mentioning of an average of 1.7 Gg/yr does not say anything about the variability of the emissions in these years (There is not a single mentioning of emissions for a specific year). It would be helpful to learn how much they varied in these years, by e.g. giving the range of emission for these few years. I presume that the numbers in parentheses are some uncertainty ranges, or are these the ranges they vary over that time?

The variations between years were not included because they are not statistically significant but we have now included them. We have also edited this section to include some more information on estimated emissions in specific years (lines 227-233).

"The modelled global annual CFC-113a emissions began in the 1960s and increased steadily at an average rate of 0.02 Gg yr$^{-1}$ yr$^{-1}$ until they reached 0.9 Gg yr$^{-1}$ (0.6-1.2 Gg yr$^{-1}$) in 2010 followed by a sharp increase to 0.52 Gg yr$^{-1}$ yr$^{-1}$ from 2010 to 2012 when emissions were 1.9 Gg yr$^{-1}$ (1.5-2.4 Gg yr$^{-1}$) (Figure 3). We find that between 2012 and 2016, modelled emissions were on average 1.7 Gg yr$^{-1}$. The best model fit (minimum-maximum) suggests some minor and statistically non-significant variability between 1.6 Gg yr$^{-1}$ (1.3-2.0 Gg yr$^{-1}$) in 2015 and 1.9 Gg yr$^{-1}$ (1.5-2.4 Gg yr$^{-1}$) in 2012. See the supplementary material for more details."

The model was run three times: once for the most likely emissions, once for the smallest possible emissions and once for the largest possible emissions. These are the values in parentheses. How the three versions of emissions were calculated is explained in the supplementary material and we have added an explanation of how the modelled uncertainties are calculated to the methods section (lines 162-166).

21. p. 5, l. 185: Can this 2% offset (using the same primary calibration scale) be related that perhaps NOAA measures the combined CFC-113a/CFC-113 isomers?

The 2% has remained approximately the same over time which means it's unlikely to be caused by combining CFC-113a and CFC-113 because their mixing ratios have been changing over time and

therefore the offset would most likely change as well. The offset is most likely due to analytical uncertainties when transferring calibration scales between labs. We decided not to include this in the paper as it is just speculation and the offset has been discussed in Laube et al., 2013.

22. Also, there does not seem to be a description anywhere on how the NOAA scale for CFC-113 was adopted, and what the propagation uncertainties of this procedures might be.

The working standard was directly purchased from NOAA and the analytical uncertainty provided by NOAA was 0.4 %. This does however not include the uncertainty of the NOAA calibration scale itself, which is currently not quantifiable as the content of CFC-113a in all standards is unknown. CFC-113a does have however a distinctly different EI mass spectrum to CFC-113, so it is unlikely that the CFC-113 used for the NOAA calibration would contain a large fraction of CFC-113a. It might however be enough to explain a 2 % offset, but the confirmation of that is outside the scope of this paper.

23. Which way is the offset, which network reports higher values?

We added more information into this sentence to say that UEA is the one with the higher mixing ratios (lines 246-248).

"There is a small offset of 2 % between the NOAA data and the current UEA Cape Grim dataset, **with the UEA Cape Grim dataset being slightly higher**, similar to the offset reported previously (Laube et al., 2013)."

24. p. 5, l. 200: It would be very informative to see a graphical comparison of the emissions for the two compounds over the last years, for example by adding those of CFC-113 to Fig 3, or to somewhere plot the ratio of the CFC-113a/CFC-113 emissions.

We agree with the reviewer and have created a new figure with all the CFC-113 emission estimates and the CFC-113a emission estimates from 1995 to 2016. This figure serves two purposes: 1) It makes the differences between the CFC-113 emission estimates for more recent years easier to see and 2) It graphically compares the CFC-113 and CFC-113a emission estimates for the last few years.

25. p. 5, l. 211: This is confusing, if the AFEAS data are used as the CFC-113 emissions in the model, then the sentence that follows does not make sense. Do the authors perhaps mean that the modeled emissions agree closely with the AFEAS bottom-up emissions?

We agree with the reviewer that the current wording is potentially misleading and have added a sentence to the text to make this clearer (lines 266-267).

"Therefore, the emissions used in our study here are the AFEAS emissions up until 1992. From 1992 onwards they are based on the best model fit to the UEA Cape Grim observations."

26. Fig. 4: colors are very hard to distinguish, for example, colors for NOAA and modelled mixing ratios seem the same. Also, filled circles don't appear as such due to their close proximities. Dashed lines appear as dash-dotted in the legend. The matching of modeled and AFEAS data is perplexing – are AFEAS data used as prior? In which year does AFEAS end, 2000; it is not possible to see this on the graph? I see only one green Rigby uncertainty line. Are the uncertainties shown for the present study based on the 'likely' or 'possible' range (p. 5 l. 22) or something else?

We agree with the reviewer and have changed the figure. The colours of the NOAA observations have been changed to a darker purple. The filled circles have been changed to lines. The legend has been changed so the emissions appear as dashes. The AFAES data is used as prior up until 1992 so they are the same as this study until then and then they diverge. This study has higher emissions after 1992 and AFEAS ends in 2003. The differences in the emissions estimates are not clear for the later years because they appear so close together on the graph. We have created a new figure with the CFC-113 and CFC-113a emission estimates for 1995-2016 to make graphical comparison clear. The lower Rigby uncertainty line is very close to this study's lower uncertainty line, which is why it is difficult to see. We have changed the emissions uncertainties to a filled area to make this clearer. We also changed the axis range of the observations to move them down slightly so they do not overlap with the emissions as much. The uncertainties for the present study are for the likely range. We have added the word "likely" to the figure caption.

27. p. 5, l. 221: Lifetime work was really done by SPARC, it might be better to cite that work.

We agree with the reviewer and have changed the reference to the SPARC 2013 report.

28. p. 5, l. 234: Do you mean 'related to CFC-113 emissions from (old) banks, i.e impurities of CFC-113a in CFC-113? This would not agree with their historic emission ratios. Please clarify.

We agree with the reviewer that this is not clear. We do not mean "related to CFC-113 emissions from old banks". We mean "related to co-emission from HFC production or agrochemical production as discussed in section 4". We have edited this sentence to be more specific (line 295).

"It should be noted that CFC-113 is not the focus of this study, but we do find that emissions of it persisted until 2017, which leaves room for the possibility that some of the recent emissions of CFC-113a are related to CFC-113 emissions, **possibly through HFC production or agrochemical production (see Section 4)** similar to findings for other isomeric CFCs (Laube et al., 2016)."

29. p. 6, l. 243: I suggest to phrase the other way: When there are spikes, then the NAME model : : :.

We agree with the reviewer and have rephrased this sentence (lines 305-308):

"In general, when there are enhancements in CFC-113a mixing ratios then the NAME footprints usually show that the air most likely came from the boundary layer over eastern China or the Korean Peninsula as shown in (a), (c), (d), and (g) for example."

30. p. 6, l. 252: Were compounds other than those listed in Table 2 also looked at?

Yes, compounds other than those listed in Table 2 were also looked at. Only the ones that we considered relevant are included in Table 2 and discussed in the text. We have edited this sentence to say that over 50 halocarbons were measured in samples from Taiwan (lines 316-319).

"After investigating correlations of CFC-113a with **over 50** other halocarbons in samples from Taiwan we found CFC-113a mixing ratios correlate well ($R^2>0.750$) in multiple years with those of CFC-113 and HCFC-133a ($CH_2ClCF_3$) indicating a possible link between the sources of these compounds (Table 2)."

31. p. 6, l. 277: 'generally': Does this mean that this was not always the case and that some Cape Grim results were higher than Tacolneston. From Fig 5 inset, it is not easy to see this.

Cape Grim is not always higher than Tacolneston. There is one data point in January 2016 when Cape Grim is higher and other measurements when the error bars overlap. The Tacolneston and Cape Grim measurements were not collected on the same days as each other so to compare them we used the dates which were closest to each other timewise.

We have changed the word in the text from "generally" to "almost exclusively (though often indistinguishable within uncertainties)", (lines 348-349).

32. Is the dark filled circle at the beginning of 2016 at _0.75 ppt from Cape Grim (with uncertainties smaller than the plotting symbol)?

The uncertainties were not smaller than the plotting symbol they just were not included. We thank the reviewer for spotting this omission. We have now changed the figure to include the uncertainties.

33. l. 279: replace 'higher' by 'lower'.

We have replaced the word "higher" with "lower".

34. p. 7, l 289: It is unclear, if only the three surface samples were used here for the interhemispheric comparison, or a historic reconstruction of the NH based on more samples. If only the surface samples were use, the mentioning of 'trend' is not adequate.

The interhemispheric comparison is from a historic reconstruction of 16 firn air samples collected at depths down to 76 metres. Only the three surface measurements were included in the figure to compare to other atmospheric measurements.

We have edited the sentence to make this clearer (lines 358-361):

"Laube et al. (2014) already found an interhemispheric gradient in CFC-113a using four of these CARIBIC flights 2009-2011 and furthermore discovered that the increasing trend of CFC-113a at Cape Grim, lagged behind the increasing trend inferred from the firn air samples, **collected to a depth of 76 metres**, from Greenland, in the Northern Hemisphere."

35. Also, Greenland firn air samples are here declared as 'representative of background Northern Hemisphere CFC-113a mixing ratio', which (similar to the same statement for Tacolneston) seems to be in contradiction to the assumption of latitudinal gradients in the NH.

We agree with the reviewer and have changed this sentence (lines 363-365):

"They will also be representative of background Northern Hemispheric CFC-113a mixing ratios **for that latitude** as the Greenland firn air location was isolated from any large industrial areas with potential sources of CFC-113a."

36. The samples are hard to see in Fig. 5, perhaps but a year in the legend for 'Greenland (surface)'.

We agree with the reviewer and have changed the legend from Greenland (surface) to Greenland 2008 (surface).

37. p. 7, l. 319: What is the 's' in parentheses?

1305    We have changed "process(s)" to "processes" (lines 399-401):

"This means that there are **processes** either producing or involving CFC-113a that lead to continuing emissions of substantial amounts of this compound, especially in East Asia."

1310    38. p. 8, l. 334: I disagree with the statement on l. 333 ff, that the absence of a correlation is not what we would expect. Before, the authors rightly state, that the correlation between CFC-113 and CFC-113a may derive from co-located factories (this is a rather likely scenario, as there are large centers of industrial activities in China). Taiwan may simply see air from places where CFC-113 and CFC-113a factories are not co-located,

1315

We agree with the reviewer and have changed this sentence to remove the part which says there the absence of a correlation is not what we would expect (lines 411-414).

"There is an absence of a correlation between CFC-113a and CFC-113 in 2015 in Taiwan and in
1320    addition, the overall mixing ratios in 2015 appear to be lower than in the other years and have fewer large enhancements (Figure 8)."

39. p. 8, l. 347: I am having difficulties to derive from Manzer 1990, that the CFC-113a route is a main pathway. It appears that he showed many pathways graphically, and listed two examples
1325    of potential pathways, neither of which was via CFC-113a. In contrast, Maranion et al., seem to suggest this route, but without backing it up with literature.

One of the pathways Manzer 1990 listed was production of HFC-134a via CFC-114a. CFC-113a can be used to make CFC-114a as shown in Manzer 1990 (Fig 3). We have changed the wording of this
1330    section so it no longer states this is one of the main routes but now states that it is one of the pathways for production of HFC-134a (line 426). We have also added in addition references in this section that also mention production of HFC-134a via CFC-113a. We also took the opportunity to add in additional references in other parts of this section to provide more references for the production methods of HFC-134a and HFC-125 and for the use of CFC-113a to produce insecticides (line 475).

1335
40. References: Laube et al., 2014: Captial 'A' For Brenninkmeijer, C. A. M.; Manzer, 1990. Reverse intials; Rigby et al., 2013: Correct Muhle

We have corrected these mistakes. The reference list is written using the Mendeley reference
1340    manager so these changes will not appear as tracked changes in the document.

41. Fig. 2: Limiting the y axes labels to the range of the data would greatly help in distinguishing the two data sets. Are these error bars now also including some uncertainties of the standard measurements, as indicated in the main text?

1345
We agree with the reviewer and have limited the y axes labels to the range of the data.

All the error bars including the ones in figure 2 are calculated the same way and include the uncertainty in the standard measurements for the day they were measured. How the error bars are
1350    calculated is described in the methods section (lines 109-111). The explanation is not included in every figure as we thought it would make all the figure captions unnecessarily long.

We have changed the methods section to explicitly say that we calculate the uncertainties the same way for all measurements.
1355

"The measurement uncertainties are calculated the same way for all measurements and represent one sigma standard deviation. They are based on the square root of the sum of the squared uncertainties from sample repeats and repeated measurements of an air standard on the same day."

1360    42. Fig. 3: Similar comment as for Fig 2 for the right y-axis: Label ticks only in range where data are. For the mean emissions, a solid line rather than a dashed line would help to see the apex better. Date 04/12/2012 not apriori clear, which is month and which is day. Legend shows light blue filled circles but these can't be seen in the plot. Suggest to replace these by lines for the model results.

1365

We agree with the reviewer and have change the axis labels, and tick marks. The mean emissions have also been changed to a solid line. We also changed the emissions uncertainties to a larger dashed line. We changed the date format to 04-Dec-2012 and we changed the filled circles to a blue line.

1370

    43. Consider to somehow show the CFC-113 emissions for the last decades in a different form, such that they could be compared to the CFC-113a emissions in Fig. 3 (perhaps add them there).

We agree with the reviewer and have created a new figure with all the CFC-113 emission estimates
1375   and the CFC-113a emission estimates from 1995 to 2016. This figure serves two purposes: 1) It makes the differences between the CFC-113 emission estimates for more recent years easier to see and 2) It graphically compares the CFC-113 and CFC-113a emission estimates for the last few years.

1380    44. Fig. 5: It is very difficult to distinguish the various sampling sets. Please improve figure.

We agree with the reviewer and have changed the figure so the data points are larger and are different shapes.

1385    45. Fig. 7: There seem to be no error bars on the CFC-113a measurements.

There are error bars they are just really small.

We have changed the colour of the error bars to a darker grey, changed the circles to no fill so the
1390   error bars underneath them can be seen and changed the axis of the CFC-113a and CFC-113 mixing ratios so the CFC-113a error bars are more stretched out.

    46. Fig. 8: Why was this particular measurement chosen? There seem to be many elevated CFC-113a shown in Fig. 5.

1395

This was an example that was representative for many other occasions. We have changed the sentence referring to this figure to explicitly say that it was representative for many other events (lines 332-333).

1400    "Figure 9 shows an example NAME footprint from a sample collected in January 2014 **that is representative for many other events.**"

    47. SI Tables: Emissions are calculated for 2017 based on 2 samples only for Jan and Feb 2017. This is almost certainly biased and I suggest to omit presenting model results for 2017.

1405

We agree with the reviewer and have removed the 2017 emissions from the figures and the supplementary material.

48. Consider complementing tables with information's on calibration scales used for the various compounds and data sets.

We agree with the reviewer and have added calibration scales to the Taiwan sheet in the spreadsheet in the supplementary material as Taiwan is the location with all the additional halocarbon measurements.

This paper updates and advances our understanding of CFC-113a in the global atmosphere, its lifetime, regions contributing emissions, and potential sources. The continued increase is interesting and important to document and understand, especially given the accelerated increase that appears to be continuing after being initially documented in an earlier paper. It is nice to see the broad range of measurement locations and information they supply. The paper is mostly sound, although there are a few sections where some reconsideration of results is warranted and where some improvement in the writing is needed. But after these issues are addressed, the paper likely would be appropriate for publication.

1. Issues to consider: As the authors note, the impact of these emissions on the ozone layer to date is minor. Suggesting that more CFC-113a might make it to the stratosphere than is indicated by surface means is a conclusion whose importance can only be speculated about (line 309-312). It is not a conclusion based on data presented here so doesn't seem appropriate to include. Geophysica results from the stratosphere are indicated as starting at background levels and decreasing above; in other words, entirely consistent with background mole fractions at Earth's surface.

We have modified this section to explicitly state that this is speculation (line 381) and have added an additional sentence (lines 385-387).

"The mechanism has already been proven to exist for shorter-lived gases (Oram et al., 2017) and we see very similar patterns transporting elevated mixing ratios of CFC-113a to the tropics very rapidly (within days) during a time of increased convective uplift."

2. Related to this, it seems important to mention in the text that a constant emission of 2 Gg/yr for a chemical with a 50-yr lifetime yields a steady-state global mole fraction of 5 ppt (15 ppt of Cl for CFC-113a). This helps the reader to objectively understand the significance of these results compared to the contribution of other chemicals including CFCs in a much more meaningful way than a comparison of cumulative emissions since 2007, for example (lines 206-209). Undoubtedly CFC-113a emissions could increase, but the potential for this is constrained by the

1455    cause of the increasing emissions and, for that fraction associated with HFC production, the
        Kigali Amendment.

        We agree with the reviewer and have included this sentence in the conclusion (lines 510-511).

1460    "For example, a constant emission of 2 Gg yr$^{-1}$ for CFC-113a yields a steady-state global mixing ratio
        of about 3.2 ppt."

        3.  Potential sources for CFC-113a emissions should be considered in light of the fact that emissions
            were fairly small until 2010, and then increased to a new value and have been essentially
1465        constant since. It is my view that this step change in emissions is primarily why this paper is
            worth publishing in ACP. It provides a strong hint as to which process likely caused this step
            change (at the least it reduces the likelihood of some causes) and is important to consider in
            gauging the likelihood of emissions increasing in the future. At the present time this section (4)
            rambles a bit and would benefit from significant tightening.
1470
        We agree with the reviewer and have edited section 4 to make it more concise.

        In the first paragraph the details of possible exceptions to the Montreal Protocol have been
        shortened. Also, the discussion of the possible causes of variations in CFC-113a and CFC-113
1475    correlations has been shortened. We have removed the example of HFC-134a in mobile air
        conditioning. We have also removed the paragraph about the differences in the correlations of CFC-
        113a and CFC-113 between Cape Grim and Taiwan. We moved the last two sentences from this
        paragraph into the paragraph about CFC-113 source banks. We removed the concluding sentence
        from the HFC paragraph and moved some of it to the concluding paragraph.
1480
        4.  On the upper lifetime limit derived for CFC-113 based on the observed rate of change of CFC-
            113. Some consideration or discussion of steady-state lifetimes vs lifetimes at zero emissions is
            required here before such a conclusion is made. These are two different quantities that have
            different values. For CFC-113, I expect its lifetime in the presence of zero emissions to be slightly
1485        shorter than one derived at SS. See papers by Prather on this topic, and consider calculating the
            difference in your model to determine if an upper limit to a SS lifetime is inconsistent or not with
            the observationally-derived value upper limit (assuming E=0).

        The SPARC Lifetime Report (Ko et al., 2013) discusses the differences between steady state lifetimes
1490    and decay times (lifetimes at zero emissions) based on the work of Prather and others. Whilst the
        decay time can differ from the steady state lifetime (Prather 1994) this difference can be very small,
        especially for long-lived gases with constant stratospheric sinks. Specifically, this difference is no
        more than 2% for gases with lifetimes greater than 10 years. CFC-113 is a long-lived gas, with a
        stratospheric sink and a likely steady state lifetime of around 82-109 years. The decay time should
1495    therefore be very similar to the steady state lifetime and any difference is relatively small compared
        to the overall uncertainty in the steady state lifetimes.

        Ko, M. K. W., Newman, P. A., Reimann, S. and Strahan, S. E., Eds.: SPARC Report on Lifetimes of
        Stratospheric Ozone-Depleting Substances, Their Replacements, and Related Species, SPARC Office.
        [online] Available from: http://www.sparc-climate.org/publications/sparc-reports/, 2013.

1500    Prather, M. J., Lifetimes and eigenstates in atmospheric chemistry, *Geophys. Res. Lett., 21,* 801-804,
        1994.

We have added the following text to section 3.1 of the paper:

"We can use the observed decrease in CFC-113 mixing ratios from 2003 onwards to calculate a decay time (lifetime at zero emissions). For long lived gases with stratospheric sinks, such as CFC-113, the decay time and steady state lifetime are very similar, differing by no more than 2 % (Ko et al., (2013). …. Accounting for the 2 % error introduced by assuming the decay time is the same as the steady state lifetime gives are overall error of 6 %. Applying this to the lifetime gives a maximum lifetime of 110 ± 7 years."

We have added the following text to section 2 of the supplementary information where we calculate the lifetime of CFC-113 by using the change in its mixing ratios at Cape Grim and a rearrangement of the chemical continuity equation:

"Accounting for the possible 2 % difference between the decay time and steady state lifetime gives an overall range of 113 ± 5 years."

5.  Line 438-440. A suggestion that new regulatory mechanisms might need to be added to the Montreal Protocol is made in the conclusion. This statement diminishes the objective nature of the data and discussion included in the paper. Policy is made with consideration of a broad range of costs and benefits, and you cannot begin to cover this complex and multifaceted discussion in a paper about atmospheric changes. If you want to comment on policy, consider doing it with an "if...then" construction. And in this case, if policy-makers wanted to require absolutely zero emissions of CFCs, then they might consider doing x,y, and z.

We agree with the reviewer and have edited this section to turn it into an 'if…then' construction (lines 524-529).

"If policy-makers wish to limit the impacts of individual isomers, then atmospheric observational data on individual CFC isomers should be reported to UNEP wherever possible. In addition, the increase in CFC-113a demonstrates that the use of ODSs as chemical feedstock or intermediates is becoming relatively more important as the use of ODSs for direct applications decreases. If policy-makers target zero emissions of CFCs, then they might consider regulating these uses of ODSs."

6.  The discussion of section 4 in the supplement is not useful without explicitly considering the changes over time in tropospheric entry values. Without this, the section adds little to the paper.

We agree with the reviewer and have added a discussion of the stratospheric mixing ratios to the supplement (lines 92-97).

"The Geophysica 2016 highest CFC-113a mixing ratio was 0.75 ± 0.02 ppt. The Tacolneston mixing ratio at this time was 0.72 ± 0.01 ppt. In 2009-2010 the Geophysica highest mixing ratio was 0.44 ± 0.01 ppt and at this time the Cape Grim mixing ratio was 0.43 ± 0.01 ppt. The highest mixing ratios observed in both campaigns agree quite well (within uncertainties) with tropospheric background mixing ratios at the time and can therefore be considered as representative of stratospheric entrance mixing ratios."

7.  Other items. How was calibration consistency maintained throughout time and across the different missions?

We have added this sentence to the methods section (lines 99-101):

1550 "All the samples are compared to the same NOAA standard (AAL-071170) and there were routine measurements of multiple standards to exclude the possibility of mixing ratio changes in the standard over time."

8.  Figure 3, mention blue solid and dashed lines in caption.

1555

We have added this sentence to the caption:

"The blue solid line represents the modelled mixing ratios with uncertainties (dashed blue line)."

1560 9.  p. 2, line 89-91, uncertainties are mentioned, but these are not the uncertainties used in the modeling, which are discussed in the supplement but not the section on modeling. I'd suggest that this appear somewhere in the main text.

We have edited and moved a section in the supplement into the 2.3 emission modelling section in
1565 the main paper (lines 162-166).

"The upper and lower emission uncertainties for CFC-113a and CFC-113 were determined by first calculating the uncertainty in matching the modelled mixing ratios with the observed mixing ratios using their recommended atmospheric lifetimes and secondly considering the uncertainty range in
1570 the lifetimes. The best fit (minimum-maximum) steady-state lifetimes used in this study are 51 years (30 years-148 years) for CFC-113a and 93 years (82 years-109 years) for CFC-113 (Ko et al., 2013; Leedham-Elvidge et al. accepted, ACP)."

10.  On uncertainties in CFC-113 calibration arising from co-elution of CFC-113a. Consider doing the
1575 atmospheric measurement community a favor by discussing the relative magnitude of interference that an analyst might have in measuring CFC-113 given co-elution of CFC-113a at the different ions these chemicals have in common (perhaps a simple table in the supplement?). This would be very helpful, and easy to add, I imagine, given that you are in a unique position to supply this important information that to first order would be independent of mass spec
1580 instrument being used.

This is unfortunately not easy to add as we are not measuring all those ions. This would also require detailed information of the setup of the different analytical systems to be compared such as GC columns, retention times, potential interferences, differences in ionisation due to different mass
1585 spectrometers (e.g. source vacuum, chemical used for mass axis calibration) and unknown quantities of CFC-113a in primary calibration standards. This would exceed the scope of this manuscript. We would however be very open to carry out a direct intercomparison experiment if approached.

11.  Lines 133-136 and 150-152. This doesn't make sense. Fitting well data at CGO wouldn't say much
1590 about the accuracy of and emission distribution in the lower SH and throughout the NH.

We agree with the reviewer and have deleted this part of the sentence:

"and has been shown to reproduce the reported mixing ratios of CFC-11 and CFC-12 at Cape Grim to
1595 within 5 % uncertainty (Reeves et al., 2005)"

12.  Lines 217, Confusing phrasing. Ultimately, global emission magnitudes derived from observations depend on the lifetime used, and you used different lifetimes than others.

1600    We agree with the reviewer and have modified this sentence (lines 270-273):

"Differences are likely due to this study using different lifetimes than Rigby et al. (2013)."

1605    13. Paragraph starting on line 251. First part: make this a discussion of variability in mole fractions and not just mole fractions. This makes your point valid and will help later when you are discussing differences in trends vs short-term variability in the UK vs Cape Grim. Second point: mention the HCFC-133a lifetime.

1610    We agree with the reviewer and have added this sentence (lines 314-316):

"The mixing ratios in Taiwan are very variable indicating nearby source region(s) whereas Cape Grim and Tacolneston mixing ratios are much less variable. Therefore, the Taiwan measurements are well suited to investigate correlations that might shed further light on potential sources."

1615
We also have added this sentence (lines 322-323):

"The tropospheric lifetime of HCFC-133a is 4-5 years (McGillen et al., 2015)"

1620
14. Para starting on line 265 (also line 291). Assertions are made that are not valid here or that extend limited results to broader context without justification (was Tacolneston sensitive to emissions from all UK source regions? Why would results from this site be representative of the NH? They might be proportional to that quantity, but not necessarily quantitatively the same).

1625    These are weakness to the paper that aren't needed and could be easily avoided.

We agree with the reviewer and have changed these sentences (lines 335-338), (lines 363-365):

"This indicates the absence of regional sources of CFC-113a in **this part of the UK**. Due to this and
1630    the relatively long lifetime of CFC-113a Tacolneston can be considered to be representative of Northern Hemisphere background mixing ratios of CFC-113a **for that latitude**."

"They will also be representative of background Northern Hemispheric CFC-113a mixing ratios **for that latitude** as the Greenland firn air location was isolated from any large industrial areas with
1635    potential sources of CFC-113a."

**List of all relevant changes**

- Line 15: author affiliation edited

1640
**Abstract**

- Line 27: "and" changed to "together with"
- Line 28: "data for" changed to "measurements in" and "lower" inserted
- Line 32: "(1.3-2.4 Gg yr⁻¹)" deleted and "(gigagrams per year)" inserted
1645    - Line 42: "substantially" moved

**1. Introduction**

- Line 45: "most of the" and "solar" inserted

- Line 47: "CFCs have negligible loss mechanisms in the troposphere and only break down when they reach the stratosphere where they are exposed to strong ultraviolet light and decompose mostly through photolysis and reaction with $O^1D$ (Ko et al., 2013)." Inserted
- Line 50: "compounds" changed to "decomposition products"
- Line 71: "IAGOS-CARIBIC project" changed to "CARIBIC-IAGOS observatory"
- Line 75: "as" inserted

- Line 78: "study" changed to "paper"

**2. Methods**

**2.1 Analytical technique**

- Line 84: "project" changed to "observatory"

- Line 85: "a glass-bottle sampler" changed to "glass bottle samplers"
- Line 85: "Various pumps were used for the different sampling activities, all of which have been thoroughly tested for a large range of trace gases (Brenninkmeijer et al., 2007; Laube et al., 2010a; Allin et al., 2015 and Oram et al., 2017)." Inserted
- Line 91: "Laube et al. (2010)" changed to "Laube et al. (2010b)"

- Line 92: "Al-PLOT" changed to "$Al_2O_3$-PLOT"
- Line 95: "A possible interference could arise when measuring CFC-113a on the GS GasPro column using m/z 116.91 if concentrations of the nearby eluding HCFC-123 are high. This was the case for a small number of samples analysed for this work and those measurements were either a) repeated using the interference-free m/z 120.90, b) replaced with measurements on the KCl-passivated $Al_2O_3$-PLOT column,

or c) excluded. The KCl-passivated $Al_2O_3$-PLOT column separated CFC-113 and CFC-113a well, no interferences were observed and m/z 116.91 was used for quantification.  All the samples are compared to the same NOAA standard (AAL-071170) and there were routine measurements of multiple standards to exclude the possibility of mixing ratio changes in the standard over time. The samples collected in Taiwan in 2013 were also measured on the Entech-Agilent GC-MS system operating in electron ionisation (EI)

mode. This consists of a preconcentration unit (Entech model 7100) connected to an Agilent 6890 GC and 5973 quadrupole MS (Leedham Elvidge et al., 2015)." Inserted
- Line 106: "On a typical day, the working standard is measured five to eight times, between every two or three samples. The sample peak sizes are measured relative to the standards measured just before and after them. The working standard is used to correct for small changes in instrument response over the course of a

day." Inserted
- Line 110: "are calculated the same way for all measurements and" inserted
- Line 111: "the" deleted and "and are derived as" changed to "They are based on"
- Line 112: "an" changed to "the"

**2.2 Sampling**

- Line 115: "This paper reports" changed to "We now report"
- Line 117: "From 2013 to 2017, 20 samples were collected at Cape Grim at irregular intervals of between one to five months apart." Inserted
- Line 118: parentheses inserted around "1978-2017"
- Line 118: "UEA Air Archive" changed to "analyses of archived air samples collected at Cape Grim,

Tasmania and analysed at the UEA, together with"
- Line 119: "in situ" italicised
- Line 120: "CFC-113 stability in the Cape Grim Air Archive has been demonstrated in the AGAGE program for periods up to 15 years and longer (Fraser et al., 1996; CSIRO unpublished data). Most of the CFC-113 UEA Cape Grim data set was previously published in Laube et al. (2013). Some of the earlier samples from

Laube et al. (2013) and Laube et al. (2014) were reanalysed on the KCl-passivated $Al_2O_3$-PLOT column (length: 50 m, ID 0.32 mm). They showed very good agreement with the previous GasPro column-based measurement with comparable precisions and no detectable offset." Inserted
- Line 128: "large-scale" changed to "extra-tropical" and "atmosphere" deleted
- Line 132: "air" inserted

1700
- Line 137: "lower" inserted, "01/09/2016 and 06/09/2016" changed to "01-Sep-2016 and 06-Sep-2016"
- Line 147: "The median mixing ratios are used for the measurements made at the Taiwan sites to decrease the influence of the large spikes in CFC-113a mixing ratios that occurred during these campaigns (Section 3.2.1). All other averages are calculated using the mean." Deleted

1705
**2.3 Emission modelling**

- Line 151: "the top down" changed to ", top down,"
- Line 152: "for the purpose of comparing the emissions of the two isomers. The model contains 12 horizontal layers each representing 2 km of the atmosphere and 24 equal-area zonally averaged latitudinal bands. The
1710 modelled mixing ratios for the latitude band that Cape Grim is located within (35.7° S–41.8° S) were matched as closely as possible to the observations at Cape Grim (40.7° S) by iteratively adjusting the global emissions rate until the differences between the modelled mixing ratios and the observations were minimised. For more details about the model see Newland et al. (2013); and Laube et al. (2016)." Inserted
- Line 159: "In this study" changed to "We now update" and "are updated" deleted
1715
- Line 161: "Air Archive" changed to "Cape Grim dataset"
- Line 163: "The upper and lower emission uncertainties for CFC-113a and CFC-113 were determined by first calculating the uncertainty in matching the modelled mixing ratios with the observed mixing ratios using their recommended atmospheric lifetimes and secondly considering the uncertainty range in the lifetimes. The 'best fit' (minimum-maximum) steady-state lifetimes used in this study are 51 years (30 years-148 years) for
1720 CFC-113a and 93 years (82 years-109 years) for CFC-113 (Ko et al., 2013; Leedham Elvidge et al., accepted ACP)." Inserted
- Line 168: "For further details see" changed to "Further details are provided in"
- Line 170: "isolated" changed to "remote"
- Line 172: "of" inserted and "is located in" deleted
1725
- Line 172: "emission" changed to "emissions"
- Line 173: "This distribution" deleted, "and" inserted
- Line 174: "and has been shown to reproduce the reported mixing ratios of CFC-11 and CFC-12 at Cape Grim within 5 % uncertainty (Reeves et al., 2005)." Deleted
- Line 175: "For CFC-113a" moved
1730
- Line 177: "for the later part of the trend" inserted
- Line 179: " for that latitude" inserted
- Line 179: "large spikes" changed to "significant enhancements"
- Line 185: "There are significant enhancements in CFC-113a mixing ratios in our measurements from Taiwan indicating continued emissions in this region (Section 3.2.1) which is consistent with emissions in this latitude
1735 band in the model." Inserted
- Line 187: "also" inserted, "our measurements from Taiwan (Section 3.2.1) and" deleted
- Line 188: "that" moved

**2.4 Dispersion modelling**

1740
- Line 210: "Dates in the NAME footprint maps are presented in the format yyyy-mm-dd and use UTC time." Inserted

**3.  Results**

**3.1 Long-term atmospheric trends and estimated global annual emissions of CFC-113a and CFC-113**

1745
- Line 214: "atmospheric" deleted, "to be" changed to "to have been"
- Line 215: "CFC-113a mixing ratios at Cape Grim" changed to "they"
- Line 216: "Overall CFC-113a mixing ratios increased gradually until about 2010 followed by a more rapid increase." Deleted

- Line 217: "CFC-113a mixing ratios increased on average by" changed to "the average rate of increase was"
- Line 218: "then" deleted, "2009" changed to "2010", "they increased by" changed to "the rate has risen threefold to"
- Line 219: "i.e. more than three times the increase from the previous period." Deleted
- Line 221: "at Tacolneston" moved, "period" inserted, "this site" changed to "it"
- Line 222: "shows" changed to "experienced"
- Line 223: "based on start and end points" inserted
- Line 223: "the median mixing ratios of CFC-113a in Taiwan increased on average by 0.06 ppt yr$^{-1}$ from 2013 to 2016. During" deleted
- Line 224: "for" inserted
- Line 226: "It has" changed to "Its atmospheric burden has"
- Line 227: "this has" inserted, "to increase" deleted
- Line 227: "therefore there must be" changed to "implying that"
- Line 228: "exceed its rate of removal" inserted, "into the atmosphere" deleted
- Line 229: "began in the 1960s and" inserted
- Line 230: "they reached 0.9 Gg yr-1 (0.6-1.2 Gg yr-1) inserted
- Line 230: "and then there was" changed to "followed by"
- Line 231: "in the average growth rate" deleted
- Line 231: "when emissions were 1.9 Gg yr$^{-1}$ (1.5-2.4 Gg yr$^{-1}$)" inserted
- Line 233: "(1.3-2.4 Gg yr$^{-1}$)" deleted
- Line 233: "(minimum-maximum)" inserted
- Line 234: "1.6 Gg yr$^{-1}$ (1.3-2.0 Gg yr$^{-1}$) in 2015 and 1.9 Gg yr$^{-1}$ (1.5-2.4 Gg yr$^{-1}$) in 2012. See the supplementary material for more details." inserted
- Line 236: "of CFC-113 at Cape Grim (Figure 4)" moved
- Line 239: "then" deleted
- Line 240: "Fraser et al., 1996" inserted
- Line 243: "CFC-113" and "CFC-113a" moved, "higher" changed to "lower"
- Line 249: "with the UEA Cape Grim dataset being slightly higher" inserted
- Line 251: "251.5" changed to "252"
- Line 252: "then" changed to "after which", "and in 2016, they are 2.4 Gg yr$^{-1}$" changed "to 2.4 Gg yr$^{-1}$ in 2016"
- Line 254: "demonstrates" changed to "witnesses"
- Line 256: "3163" changed to "3164"
- Line 258: "37" changed to "38"
- Line 259: "which is about 35%" changed to "or a third"
- Line 259: "over this period" deleted
- Line 260: "This indicates that as emissions of other CFCs are decreasing CFC-113a becomes relatively more important." Deleted
- Line 260: "Current" inserted
- Line 261: "CFC-113a emissions are now similar to CFC-113 emissions and could even become larger in the future if the current trends continue." Changed to "Current CFC-113a emissions are similar to those of CFC-113 and could even surpass them if the trends continue (Figure 5).
- Line 265: "In the model, these emissions lead to a best-fit match to the CFC-113 observations." Inserted
- Line 266: "AFEAS reports were" changed to "AFEAS report data"
- Line 269: "Therefore, the emissions used in our study here are the AFEAS emissions up until 1992. From 1992 onwards they are based on the best model fit to the UEA Cape Grim observations." Inserted
- Line 271: "These" changed to "Those", "the emissions estimated in this study" changed to "ours"
- Line 272: "and the differences are likely due to differences in the methods used to calculate the emissions. Rigby et al. (2013) used the estimated emissions to derive the steady state atmospheric lifetimes whereas in this study, we used the steady state atmospheric lifetimes to derive the emissions using shorter lifetimes than in Rigby et al. (2013)" changed to "Differences are likely due to this study using different lifetimes than Rigby et al. (2013)."
- Line 278: "WMO" changed to "SPARC", "(Carpenter and Reimann, 2014)" changed to "(Ko et al., 2013)

- Line 279: "(Carpenter and Reimann, 2014)" changed to "(Ko et al., 2013)
- Line 280: "with" changed to "using"
- Line 280: "quickly enough" changed to "sufficiently rapidly"
- Line 281: "observed" inserted, "observations" deleted, "with no emissions" changed to "in the absence of emissions"
- Line 282: "This indicates that the maximum possible lifetime of CFC-113 is somewhere between 109 years and 138 years." Deleted
- Line 283: "We can use the observed decrease in CFC-113 mixing ratios from 2003 onwards to calculate a decay time (lifetime at zero emissions). For long lived gases with stratospheric sinks, such as CFC-113, the decay time and steady state lifetime are very similar differing by no more than 2 % (Ko et al., 2013)." Inserted
- Line 286: "reproduced" changed to "reproduces"
- Line 287: "maximum" deleted
- Line 287: "By assuming zero emissions, this lifetime is a maximum value, since any source of CFC-113 would have to be balanced by a shorter lifetime." Inserted
- Line 289: "Accounting for the 2 % error introduced by assuming the decay time is the same as the steady state lifetime gives are overall error of 6 %." Inserted
- Line 291: "110 ± 6" changed to "110 ± 7"
- Line 294: "113 ± 4" changed to "113 ± 5"
- Line 295: "enables" changed to "leaves room for"
- Line 297: "possibly through HFC production or agrochemical production (see Section 4)" inserted

**3.2 Global distributions of CFC-113a**

**3.2.1 Enhancement above background mixing ratios**

- Line 301: "at Taiwan" changed to "observed in Taiwan", "light blue dots" changed to "light blue stars", "Figure 5" changed to "Figure 6"
- Line 302: "considered" inserted
- Line 303: "ppt" deleted
- Line 304: "large spikes" changed to "enhancements"
- Line 205: "most likely continental East Asia" inserted
- Line 306: "These enhancements in CFC-113a mixing ratios are likely due to emissions of this compound in East Asia." Deleted
- Line 307: "origin" changed to "emissions", "are" changed to "were", "Figure 6" changed to "Figure 7"
- Line 308: "when there are" inserted, "the spikes" changed to "enhancements", "then the" inserted, "usually occur when the" deleted
- Line 309: "usually" inserted
- Line 310: "peninsula" changed to "Peninsula", "as shown" inserted, "Whereas" changed to "In contrast"
- Line 311: "correspondingly" inserted
- Line 312: "peninsula" changed to "Peninsula"
- Line 312: "In (b) and (e), the air is coming from the north, between the eastern coast of China and the Korean peninsula, and in (f) the air mass originates predominantly from over the Pacific Ocean." Deleted
- Line 316: "The mixing ratios in Taiwan are very variable indicating nearby source region(s) whereas Cape Grim and Tacolneston mixing ratios are much less variable. Therefore, the Taiwan measurements are better suited to investigate correlations that might shed further light on potential sources." Inserted
- Line 319: "a range of" changed to "over 50", "samples from" inserted
- Line 321: "There is a great deal of variability in mixing ratios in the Taiwan samples." Inserted
- Line 323: "Figure 7" changed to "Figure 8"
- Line 324: "The tropospheric lifetime of HCFC-133a is 4-5 years (McGillen et al., 2015)" inserted
- Line 325: "HCFC-133a" changed to "and its"
- Line 325: "over the last few years" changed to "in recent years"
- Line 326: "then the trend reversed and" changed to "decreased", "they decreased" deleted
- Line 327: "Then" deleted

- Line 328: "this" changed to "such a"
- Line 328: "it is currently unclear what is causing these changes" changed to "the causes of these changes are still unclear"
- Line 330: "from" changed to "at", "grey dots" changed to "grey crosses", "Figure 5" changed to "Figure 6"
- Line 331: "as much" changed "to the same degree"
- Line 331: "The mixing ratios" changed to "they"
- Line 332: "East Asia" changed to "East Asian air masses"
- Line 334: "Figure 8" changed to "Figure 9"
- Line 335: "that is representative for many other events" inserted
- Line 336: "yellow dots" changed to "yellow diamonds", "Figure 5" changed to "Figure 6", "large spikes" changed to "significant enhancements"
- Line 337: "that Tacolneston is not located close to any sources of CFC-113a and therefore" changed to "the absence of regional sources"
- Line 338: "in this part of" inserted, "does not have large sources of CFC-113a" deleted
- Line 340: "for that latitude" inserted
- Line 341: "large spikes" changed to "many enhancements"
- Line 343: "24/03/16" changed to "24-Mar-2016
- Line 344: "Figure 6" changed to "Figure 7", "04/04/16" changed to "04-Apr-2016"
- Line 347: "suggesting" changed to "indicating"

**3.2.2 Interhemispheric gradient of CFC-113a**

- Line 350: "generally higher" changed to "almost exclusively higher (though often indistinguishable within uncertainties)"
- Line 351: "Figure 5" changed to "Figure 6"
- Line 352: "higher" changed to "lower"
- Line 354: "interhemispheric" deleted
- Line 355: "data from" inserted, "from Germany to South Africa between 2009 and 2016" changed to "between Germany and South Africa for 2009-2016"
- Line 356: "purple dots" changed to "purple circles, Figure 6"
- Line 357: "Fig. 5" changed to "Fig.6"
- Line 358: "dots" changed to "circles"
- Line 359: "dots" changed to "circles"
- Line 360: "also" changed to "already"
- Line 361: "they found" changed to "discovered"
- Line 362: "collected to a depth of 76 metres" inserted
- Line 366: "for that latitude" inserted
- Line 367: "Figure 5" changed to "Figure 6", "dots" changed to "crosses"
- Line 368: "dot" changed to "cross", "figure" changed to "Figure"
- Line 372: "or" changed to "with"

**3.2.3 Measurements of CFC-113a in the stratosphere**

- Line 377: "Nearly all" inserted, "are usually" deleted, "on the" changed to "during", "at" changed to "represent"
- Line 379: "dots" changed to "diamonds", "Figure 5" changed to "Figure 6", "at" changed to "near"
- Line 379: "The measurements should thus be representative of the mixing ratios of compounds just" changed to "Their mixing ratios should be representative for air masses"
- Line 381: "layer" changed to "region"
- Line 382: "9/11/2013" changed to "9-Nov-2013"
- Line 383: "Figure 5" changed to "Figure 6", "We speculate that" inserted

- Line 387: "The mechanism has already been proven to exist for shorter-lived gases (Oram et al., 2017) and we see very similar patterns transporting elevated mixing ratios of CFC-113a to the tropics very rapidly (within days) during a time of increased convective uplift." Inserted
- Line 391: "take samples of stratospheric air" changed to "sample lower stratospheric air"
- Line 392: "dots" changed to "squares"
- Line 393: "Figure 5" changed to "Figure 6"
- Line 396: "mainly" inserted, "This is" deleted

**4. Possible sources of CFC-113a**

- Line 401: "It is therefore likely that there is a continuing industrial process(s)" changed to "This means that there are processes"
- Line 402: "leads" changed to "lead"
- Line 403: "into the atmosphere" deleted
- Line 405: "chemical intermediates and fugitive emissions" inserted
- Line 405: "Whilst feedstock usage has to be reported to the United Nations Environment Programme (UNEP), these data are not publicly available" deleted
- Line 406: "Also" changed to "As"
- Line 407: "so" deleted
- Line 408: "Furthermore, the use of ODSs as intermediate species and trace amounts of fugitive emissions do not have to be reported. Therefore, possible sources for the increase in CFC-113a mixing ratios include its use as a chemical feedstock, chemical intermediates, and fugitive emissions as well as unsanctioned production (Laube et al., 2014)." Deleted
- Line 413: "The absence" changed to "There is an absence"
- Line 414: "is not what we would expect based on their source type (industry) and lifetimes" deleted
- Line 416: "spikes changed to "enhancements", "Figure 7" changed to "Figure 8"
- Line 418: "China/Korea were the areas" changed to "Regions in China and Korea"
- Line 419: "source" changed to "locations"
- Line 420: "indicate that there is more than one process emitting CFC-113a in East Asia, or variability in the process or in the amount of leaking of gases. This may" deleted
- Line 424: "this" changed to "it"
- Line 427: "may" inserted, "There are two main routes for making HFC-134a (Manzer, 1990)" deleted
- Line 428: "One of the pathways for production of HFC-134a" inserted, "The first" deleted
- Line 430: "Rao et al., 1992; Bozorgzadeh et al., 2001; Maranion et al., 2017" inserted
- Line 431: "The other" changed to "Another"
- Line 436: "any" deleted, "was practiced" inserted
- Line 439: "no" changed to "not"
- Line 446: "Observed" deleted
- Line 449: "For example, as HFC-134a is used in mobile air conditioning units and in refrigeration, we would expect a significant component of HFC-134a emissions to be related to automobile use." Deleted
- Line 458: "On the one hand, the Cape Grim dataset shows that CFC-113a and CFC-113 have very different atmospheric trends (Figures 3, 4) but on the other hand, the Taiwan dataset shows that the isomers are mostly strongly correlated (Figure 7). This is not necessarily a contradiction because, close to sources the two compounds would still be correlated, but the emissions are low in absolute numbers for CFC-113 so it is globally still decreasing. In Sect. 3.1 we concluded that there was possibly a small amount of continued emissions of CFC-113 to maintain the observed atmospheric mixing ratios. This would be consistent with either a source from banks and/or release in conjunction with CFC-113a." deleted
- Line 465: "and/" inserted, "peninsula" changed to "Peninsula", "sources of" changed to "source regions for"
- Line 471: "and as we do not know where they are located" deleted
- Line 472: "this" changed to "which"
- Line 472: We suggest that the CFC-113a emissions in the atmosphere originate predominantly from HFC production; however, there is currently insufficient data available to conclude this with high confidence.
- Line 477: "Jackson et al., 2001; Cuzzato and Bragante, 2002" inserted
- Line 478: "In addition" moved

- Line 486: "In Sect. 3.1 we concluded that there was possibly a small amount of continued emissions of CFC-113 to maintain the observed atmospheric mixing ratios. This would be consistent with either a source from banks and/or release in conjunction with CFC-113a."
- Line 492: "however; there is currently insufficient data available to conclude this with high confidence."

**5. Conclusions**

- Line 508: "declining" inserted
- Line 509: "yet" deleted
- Line 510: "major" deleted
- Line 512: "For example, a constant emission of 2 Gg yr$^{-1}$ for CFC-113a yields a steady-state global mixing ratio of about 3.2 ppt." inserted
- Line 517: "and whilst CFC-113a emissions have appeared to be stable in recent years this does not mean that they will not increase in the future." Inserted
- Line 520: "What is required is" changed to "When"
- Line 521: "become available" inserted, "With such data" deleted
- Line 522: "could" changed to "can"
- Line 526: "Therefore, we recommend that" changed to "If policy-makers wish to limit the impacts of individual isomers, then"
- Line 527: "should" inserted
- Line 529: "ODSs used as chemical feedstock or intermediates might need to start being regulated by the Montreal Protocol as the use of ODSs for direct applications decreases, the use of ODSs as chemical feedstock or intermediates is becoming relatively more important." Changed to "the use of ODSs as chemical feedstock or intermediates is becoming relatively more important as the use of ODSs for direct applications decreases. If policy-makers target zero emissions of CFCs, then they might consider regulating these uses of ODSs."

**Acknowledgements**

- Line 539: "(CARIBIC-IAGOS)" inserted

**Figures and Tables**

- Table 1: all dates changed to dd-mmm-yyyy format
- Figure 2: axis and tick marks changed and 2017 emissions removed
- Figure 3: axis and tick marks changed, the modelled mixing ratios changed from filled circles to a line, the 'best-fit' emissions changed from dashed line to a solid line, 2017 emissions and modelled mixing ratios removed, in the figure caption "04/12/2012" changed to "04-Dec-2012", "The blue solid line represents the modelled mixing ratios with uncertainties (dashed blue line)." Inserted
- Figure 4: NOAA observations colour changed, filled circles changed to lines, emissions in legend changed to dashes, emissions uncertainties changed to a filled area, axis range of the observations changed, 2017 emissions and modelled mixing ratios removed and "likely" inserted into the figure caption
- Figure 5: new figure, caption: "CFC-113 emissions from this study, AFEAS and Rigby et al 2013 and CFC-113a emissions from this study 1995-2016 with uncertainties."
- Figures after Figure 5 renumbered
- Figure 6: error bars added, size and shape of data points changed
- Figure 7: all dates changed to dd-mmm-yyyy format
- Figure 8: axis changed, colour of error bars changed, colour of data points changed, all dates changed to dd-mmm-yyyy format
- Figure 9: date changed to dd-mmm-yyyy format